# Supervised Classification Heads as Semantic Prototypes: Unlocking Vision-Language Alignment via Weight Recycling

**David Méndez** [1]   **Roberto Confalonieri** [2]   **Natalia Díaz-Rodríguez** [1]

## Abstract

Vision-Language Models (VLMs) excel at tasks like zero-shot classification and cross-modal retrieval by mapping images and text to a shared space, but this requires expensive end-to-end training with massive paired datasets. Current post-hoc alignment methods reduce computational costs by connecting pretrained encoders through lightweight mappings, yet still demand substantial paired data. In this work, we investigate the potential of repurposing the classification heads of pretrained vision models as *semantic prototypes*. The recycling of these weights, typically discarded after pretraining, unlocks two distinct capabilities: it enables zero-shot alignment by using weights as semantic anchors, and serves as a robust data augmentation strategy by mixing these prototypes with real image-text pairs. We demonstrate that integrating our approach with several state-of-the-art post-hoc alignment techniques consistently boosts accuracy in cross-modal retrieval, zero- and few-shot classification tasks.

## 1. Introduction

Vision-language models have demonstrated powerful capabilities—such as zero-shot classification, cross-modal retrieval, image captioning, and visual question answering—by embedding images and text into a shared representation space. Classic approaches (Radford et al., 2021; Jia et al., 2021) achieve this by jointly training image and text encoders on massive paired datasets. However, collecting and curating billions of image-caption pairs and then optimizing two large models end-to-end incurs prohibitive

data and computational costs for most researchers and institutions, effectively restricting the training of such models to those with considerable resources. Recent *decoupled* techniques (Zhai et al., 2022; Cao et al., 2025) freeze an encoder that is pretrained on one modality and then rely on contrastive training to align the other modality's encoder to this fixed representation, still requiring substantial paired data and computation. Post-hoc alignment methods (Norelli et al., 2023; Li et al., 2025; Maniparambil et al., 2024) learn lightweight functions that map representations from independently pretrained encoders to a common latent space, eliminating large-scale contrastive training but still relying on extensive paired image-text data to achieve vision-language alignment.

We study whether semantic information already present in supervised vision models can be reused for vision-language alignment. We demonstrate that the classification head weights of vision models pretrained on ImageNet-21K (Ridnik et al., 2021), which are typically discarded after pretraining, function as high-quality *semantic prototypes*. By repurposing classification head weights as semantic prototypes to align an image feature extractor to a text encoder, we enable vision-language capabilities in two regimes: alignment without any image-text paired data, and a data augmentation strategy where these prototypes are combined with real image-text pairs thanks to their intrinsic compatibility. In this post-hoc alignment setting, pretrained encoders are treated as fixed and the focus is on reducing the additional paired data needed to connect them. Our results show promising performance in cross-modal retrieval and classification tasks. In brief, we highlight our key contributions here:

- **Classification heads as semantic prototypes.** We provide empirical validation and theoretical justification for treating classification head weights as meaningful semantic prototypes of the classes they are associated with. We show that the weights for a class align with text representations even better than the average of images belonging to that class.

- **Vision-language alignment without image-text pairs.** Leveraging the previous insight, we introduce a weight

---

[1]Department of Computer Science and Artificial Intelligence, DaSCI Institute, University of Granada, Granada, Spain [2]Department of Mathematics "Tullio Levi-Civita", University of Padova, Padova, Italy.
Correspondence to: David Méndez <davidmendez@ugr.es>.

*Proceedings of the $43^{rd}$ International Conference on Machine Learning*, Seoul, South Korea. PMLR 306, 2026. Copyright 2026 by the author(s).

recycling framework to align independently trained image and text encoders in a post-hoc manner without any image-text pairs.

- **Compatibility with real image-text pairs.** We demonstrate that classification head weights are highly compatible with real image-text representations. Beyond enabling alignment in the absence of paired data, they serve as a robust, complementary data augmentation source that consistently boosts the performance of state-of-the-art post-hoc methods when combined with available image-text pairs.

- **Analysis of weight representations.** We provide geometric and domain-specific analyses of classification head weights, including their relationship to downstream tasks and their separation from image representations.

This work offers a new perspective on the emergence of semantics in deep vision models pretrained at scale with supervision, paving the way for alignment strategies that recycle supervised pretraining weights.

## 2. Related Work

**Vision-Language Models trained from scratch.** Pioneering vision-language models like CLIP (Radford et al., 2021) or ALIGN (Jia et al., 2021) jointly train vision and text encoders with a contrastive loss, so that an image and a text are mapped to similar representations if and only if they share the same semantic content. The resulting aligned vision-language space enables zero-shot transfer across downstream tasks without requiring task-specific fine-tuning. More recent approaches demonstrate that powerful off-the-shelf encoders can be leveraged to avoid training both modality encoders. LiT (Zhai et al., 2022) freezes a pretrained vision backbone and trains only the text encoder, while FLAME (Cao et al., 2025) does the opposite–freezing the text model and learning the parameters of the vision encoder. However, these approaches still require substantial computational costs, as they involve training a large encoder with hundreds of millions of parameters.

**Post-hoc vision-language alignment with image-text pairs.** A growing body of work focuses on connecting image and text encoders that were not jointly trained. The text-to-concepts framework (Moayeri et al., 2023), for example, starts with a pretrained vision-only encoder and learns a single-layer MLP that maps its image representations into CLIP's latent space. The combination of the vision encoder and the MLP, together with CLIP's text encoder, produces similar representations for images and texts that share the same semantic content, in a manner analogous to CLIP's

original image and text encoders. Additionally, some methods aim to align image and text encoders that have been trained each exclusively on unimodal data. ASIF (Norelli et al., 2023) embeds visual and text data into a common space using a fixed set of image-caption pairs: for a given image, its representation is the vector whose $i$-th component is the similarity (in the original image encoder's space) between that image and the $i$-th image in the fixed set; likewise, for a given text, its representation is the vector whose $i$-th component is the similarity (in the original text encoder's space) between that text and the $i$-th caption in the fixed set. CSA (Li et al., 2025) applies Canonical Correlation Analysis to project pretrained image and text embeddings into a maximal correlation subspace. SAIL (Zhang et al., 2025) contrastively trains two linear transformations on millions of image-caption pairs to align representations from both modalities. The QPA method (Maniparambil et al., 2024) addresses alignment as a seeded graph-matching problem: it formulates a Fast Quadratic Assignment Problem optimization that directly aligns the similarity graphs of the two modalities. The issue with these approaches is that they require a sufficiently large and diverse set of multimodal image-text pairs for the alignment to work effectively, which may not be available in many practical domains and thus limits their applicability.

**Zero-shot classification by mapping text to weights.** A related line of work creates zero-shot classifiers by mapping text representations directly to the weights of image classifiers. For example, methods like ICIS (Christensen et al., 2023) and Zero-Shot Natural Language Explanations (Sammani & Deligiannis, 2025) use lightweight MLPs to transform text embeddings into classification weights. These approaches are constrained to the ImageNet-1K or other more specific domains, limiting their effectiveness beyond these settings.

While previous work focuses on either text-to-image alignment or mapping text to new weights for zero-shot classification, we take a conceptually different approach by leveraging the semantic power of usually discarded classification head weights. Specifically, we leverage the larger-scale classification head weights from ImageNet-21K pretraining—despite being noisier than other pretraining sets such as ImageNet-1K—in two distinct ways: they either enable vision-language alignment without any paired data or serve as robust augmentation data for alignment when combined with real image-text pairs. We note that the data cost of endowing a vision encoder with language capabilities is different from the (potentially large) datasets used to pre-train the image and text encoder components, which we treat as fixed, off-the-shelf models.

## 3. Supervised Pretraining Classification Heads as Semantic Prototypes

Let $\mathcal{X}$ denote the input image space and $f_I : \mathcal{X} \longrightarrow \mathbb{R}^d$ the image encoder mapping each image into a $d$-dimensional feature vector. We assume the image encoder $f_I$ undergoes a pretraining phase, in which a linear classifier is optimized on top of $f_I$:

$$W f_I(x) + b \qquad (1)$$

where $W \in \mathbb{R}^{C \times d}$, with $C$ denoting the number of classes (e.g., 21,841 in ImageNet-21K), and $b \in \mathbb{R}^C$ is the bias term. During pretraining, $W$, $b$, and $f_I$ are jointly optimized; afterwards, $f_I$ serves as frozen feature extractor.

**Weight vectors encode semantic prototypes.** Our core insight is that each row $w_i$ of the weight matrix $W$ in Equation (1) functions as a latent prototype of its associated semantic concept. To validate this, we extract the $w_i$ vectors corresponding to ImageNet-1K classes from a model pretrained on ImageNet-21K and compare two classifiers on the ImageNet-1K validation set: (i) the standard linear classifier (dot product + bias), and (ii) a cosine-similarity classifier using $\cos(w_i, f_I(x))$. We conduct this experiment across five diverse architectures, including transformers (BEiT, TinyViT), CNNs (ConvNeXt), and hybrids (CAFormer, ConvFormer). As shown in Table 1, the cosine classifier consistently maintains high accuracy (above 80% across all architectures) comparable to the linear baseline. This empirically demonstrates that the directional information in $w_i$ encodes a robust representation of the class concept, independent of the bias term or vector magnitude.

**Theoretical framing: Neural Collapse.** The Neural Collapse phenomenon (Papyan et al., 2020) could be argued to provide theoretical insight into why classifier weight rows serve as class prototypes. This phenomenon describes the convergence of neural network parameters to a geometric configuration with self-duality: the row vector $w_i$ of $W$ and the mean of class $i$ representations converge up to rescaling. As a consequence, in the training limit, $w_i$ becomes the (scaled) class prototype (see Section B in Appendix for mathematical discussion). We must remark, however, that Neural Collapse has been studied under terminal phase training with zero-error, number of classes not exceeding feature dimension, and balanced class distributions. While recent

extensions address some of these limitations individually by adding new technical conditions (Jiang et al., 2024; Yang et al., 2022), to the best of our knowledge, there are still gaps between theoretical assumptions and the large-scale pretraining that optimizes the parameters in Equation (1). Hence, we examined the cosine classifier accuracy Table 1 as direct evidence that the $w_i$ vectors encode meaningful class representations, even when the conditions for Neural Collapse are not fully satisfied.

**Comparison of Semantic Alignment: Head weights vs Averaged Image Representations.** Beyond justifying why weight rows can serve as class prototypes, we investigate how well they align with *language*. We compare the alignment of (a) weight vectors $w_i$ and (b) averaged image embeddings against the text embeddings of their class names. We employ the mutual k-nearest-neighbor alignment metric ($m_{NN}$) (Huh et al., 2024), which measures the preservation of semantic neighborhoods across modalities. This metric measures how well two spaces preserve semantic neighborhoods. We extract the 1,000 row vectors $w_i$s from the ImageNet-21K classification head matrix that correspond to ImageNet-1K classes and compute their $m_{NN}$ alignment scores with the text embeddings of the corresponding class names. In parallel, we compute $m_{NN}$ scores for averaged image embeddings from the ImageNet-1K validation set, where we vary the number of images $n$ per class used in each average ($n \in \{1, 5, 10, 20, 30, 40, 50\}$, with 50 representing the full validation set per class). Figure 1 shows that row vectors from the classification head matrix consistently achieve higher $m_{NN}$ scores than averaged image embeddings across all values of $n$ and neighborhood sizes $k \in \{3, 5, 10\}$. This suggests that classification heads exhibit stronger semantic alignment with text embeddings than the average of visual features, making $W$ a superior anchor for vision-language alignment.

Overall, these findings establish classification head weights as high-quality semantic prototypes that are intrinsically aligned with language.

## 4. Unlocking Vision-Language Alignment via Weight Recycling

Building upon the insight that classification head weights, typically discarded after large-scale supervised pretraining, can be leveraged as semantic prototypes, we propose a framework to repurpose them for vision-language alignment. We integrate these classification heads into post-hoc alignment methodologies (Figure 2) to achieve two main objectives: first, to unlock vision-language capabilities in the complete absence of paired data (alignment without image-text pairs); and second, to demonstrate that these weight representations are compatible with real image-text pairs,

*Table 1.* Accuracy (%) on ImageNet-1K validation set for a linear layer $W f_I(x) + b$ and cosine similarity $\cos(w_i, f_I(x))$, where $w_i$ is $W$'s i-th row. The high accuracy in the cosine classifier indicates the $w_i$s are meaningful representations of their classes, for all encoders.

| Last layer | BEiT | CAFormer | ConvFormer | ConvNeXt | TinyViT |
|---|---|---|---|---|---|
| $W f_I(x) + b$ | 82.71 | 80.81 | 80.76 | 83.29 | 82.31 |
| $\cos(w_i, f_I(x))$ | 82.10 | 80.28 | 80.08 | 82.67 | 81.30 |

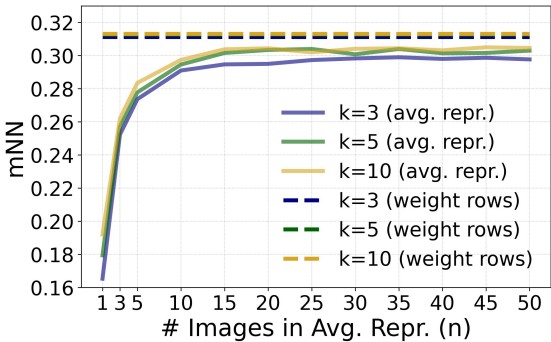

*Figure 1.* Mutual k-NN alignment ($m_{NN}$) to text representations for classification head vectors (dashed) and averaged image embeddings (solid) as a function of the number of images per class $n$. Multiple neighbors $k \in \{3, 5, 10\}$ are tested. Representations are computed using the non-aligned BEiT-B/16 (Bao et al., 2021) image encoder and CLIP's text encoder. Across all tested $k$ and $n$, classification heads exhibit higher alignment to the text representations than image representation averages.

allowing them to be combined to boost the performance of existing post-hoc alignment methods. In both regimes, we evaluate the resulting aligned models on classification and cross-modal retrieval downstream tasks. We begin by formulating the post-hoc alignment problem.

**General post-hoc alignment setting.** By post-hoc alignment, we refer to methods that learn lightweight functions to align independently pretrained encoders, without modifying the encoder parameters. We formulate this task mathematically. Let $\mathcal{X}$ denote the input image space and $\mathcal{T}$ denote the input text space. We define our image and text encoders as:

$$f_I : \mathcal{X} \longrightarrow \mathbb{R}^d \qquad \text{and} \qquad f_T : \mathcal{T} \longrightarrow \mathbb{R}^{\overline{d}},$$

where $f_I$ maps each image into a $d$-dimensional feature vector and $f_T$ maps text inputs into $\overline{d}$-dimensional feature vectors. If $f_I$ and $f_T$ had been jointly optimized with a contrastive loss as in CLIP (Radford et al., 2021), then $d = \overline{d}$ and they would satisfy

$$f_I(x) \approx f_T(t)$$

for image-text pairs $(x, t)$ with the same semantic content (i.e. an image and a textual description of it), while pushing apart pairs with different semantic content. In the post-hoc alignment setting, we assume independently pretrained encoders, thus $f_I$ and $f_T$ are not aligned. To bridge this gap, the objective is that lightweight functions $g : \mathbb{R}^d \to \mathbb{R}^s$ and $\overline{g} : \mathbb{R}^{\overline{d}} \to \mathbb{R}^s$ are learned to map each modality's space into a common space $\mathbb{R}^s$ of dimension $s$, such that for any image-text pair $(x, t)$ sharing the same semantic content,

$$g(f_I(x)) \approx \overline{g}(f_T(t)) \qquad (2)$$

To learn $g$ and $\overline{g}$, an alignment dataset

$$\widetilde{\mathcal{D}}_{imgtxt} = \{(x_j, t_j)\}_{j=1}^p \subseteq \mathcal{X} \times \mathcal{T}$$

with $p$ image-text pairs is used. Since $g$ and $\overline{g}$ act on the image and text representation spaces respectively, we only need the representations of image-text pairs in $\mathcal{D}_{imgtxt}$ to learn $g$ and $\overline{g}$. Thus, we define a dataset $\mathcal{D}_{imgtxt}$ as

$$\mathcal{D}_{imgtxt} = \{(f_I(x_j), f_T(t_j))\}_{j=1}^p$$

and focus on it instead of on the original image-text alignment dataset $\widetilde{\mathcal{D}}_{imgtxt}$. In the case of CSA (Li et al., 2025) and SAIL (Zhang et al., 2025), $g$ and $\overline{g}$ are linear projections learned using Canonical Correlation analysis and contrastive learning respectively. ASIF (Norelli et al., 2023) uses a fixed set $\mathcal{D}_{imgtxt}$ of image-text representations and defines $g$ as the function taking an image representation to a vector of similarities with respect to the vision part of $\mathcal{D}_{imgtxt}$ and analogous with $\overline{g}$ and text representations. In contrast to ASIF and CSA, the text-to-concepts approach (Moayeri et al., 2023) takes a different strategy. Instead of learning two lightweight mappings that project both modalities into a third representation space, it learns either $\overline{g}$ or $g$ to project one modality's representation space into the other modality's representation.

In the rest of this section, we show how classification heads can be recycled to both unlock language capabilities in vision-only backbones without paired data and augment real image-text pairs to boost alignment performance. We do so by leveraging post-hoc alignment methods as an application of our central finding: classifier weights discarded after pretraining constitute valuable reusable semantic anchors. This application is most relevant when a specific supervised vision backbone should be retained, for example because of deployment constraints, computational efficiency, architectural requirements, or compatibility with existing interpretability tools. Throughout these experiments, the image and text encoders are treated as fixed off-the-shelf components, therefore, any discussion of data and computational cost discussion refers only to the additional alignment step.

In addition to two state-of-the-art post-hoc alignment methods, such as CSA (Li et al., 2025) and text-to-concepts (Moayeri et al., 2023), operating in a low data regime, we employ a strong baseline that trains a lightweight MLP to map text representations to image representations, which we refer to as MLP-alignment. This approach—using a lightweight MLP trained with image-text pairs to align vision and language encoders—has been widely adopted in the literature, dating back to early image-text alignment works such as (Frome et al., 2013). In particular, we choose a two-layer MLP, as this proved to yield better results than a single-layer MLP, and train it with cosine-similarity loss.

Once the vision backbone has been endowed with language capabilities via post-hoc alignment, we evaluate its performance on classification benchmarks and cross-modal

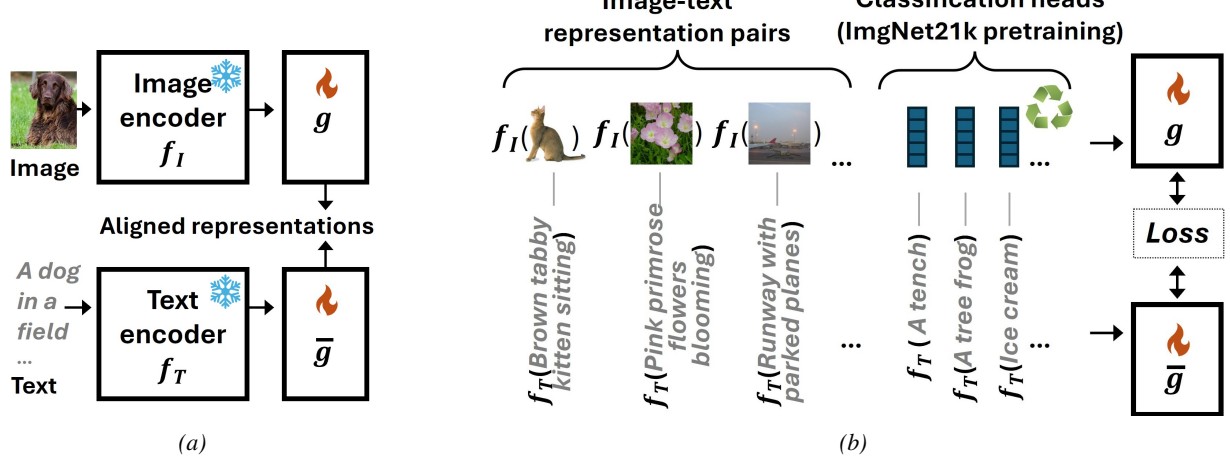

*Figure 2.* Approach to leverage classification heads in post-hoc representation alignment. (a) Illustration of the post-hoc alignment setting, in which lightweight functions $g$ and $\overline{g}$ are learned to map representations from independently trained image and text encoder to an image-text aligned space. (b) We recycle the classification head weights from ImageNet-21K pretraining with their class names to augment image-text data for learning $g$ and $\overline{g}$. Recycled weights can also be used alone to unlock alignment without image-text pairs.

retrieval tasks (image-to-text and text-to-image retrieval). Throughout this section, we use the BEiT-B/16 (Bao et al., 2021) image encoder and CLIP's ViT-B/32 text encoder (Radford et al., 2021) as our image and text encoder respectively. Further experiments on tasks using additional image and text encoders—also those trained solely on text, such as RoBERTa (Liu et al., 2019) or MPNET (Song et al., 2020)—along with technical details (e.g., hyperparameters, hardware, training time) and code are provided in the Appendix. As a remark, we note that learning the $g$ and $\overline{g}$ functions in our experiments is computationally inexpensive, requiring less than 2 minutes on a single GPU.

### 4.1. Vision-language alignment with no image-text pairs.

As an application of the insight that classification head weights serve as semantic prototypes, we show how they can be leveraged to enable post-hoc vision-language alignment in the complete absence of paired image-text data, i.e., when $\mathcal{D}_{imgtxt} = \emptyset$. In such a case, we recycle the row vectors $w_i$ of $W$ as semantic prototypes and pair them with the text embeddings of their corresponding class names. Formally, we define the dataset

$$\mathcal{D}_{weights} = \{(w_i, f_T(t_i))\}_{i=1}^{C}$$

where $t_i$ is the name of the i-th class in the pretraining dataset (e.g., 'A tench', 'A tree frog', etc.). We then use $\mathcal{D}_{weights}$ to learn the alignment functions $g$ and $\overline{g}$ as in Equation (2). This approach effectively aligns the image and text encoders without requiring any paired image-text data, unlocking zero-shot vision-language capabilities. We assess the performance of the resulting aligned models on zero-shot classification and cross-modal retrieval.

**Zero-shot classification.** In Figure 3, we report the zero-shot classification accuracy of the MLP-aligned model using weight-only data, comparing results obtained from ImageNet-1K and ImageNet-21K pre-training weights. Although ImageNet-21K classification head weights are noisier due to class hierarchy complexity, they provide representations for significantly more classes than ImageNet-1K, which appears to be the key factor driving performance. The figure also includes results where we combine classification weights with a few image-text pairs, which further improves performance as discussed in Section 4.2.

**Cross-modal retrieval.** Cross-modal retrieval results using MLP, CSA, and text-to-concepts methods are reported in Appendix Table 4, demonstrating that weight-only alignment enables retrieval capabilities significantly above random chance. Our experiments show that even without any paired image-text data, leveraging classification head weights enables meaningful vision-language alignment and yields competitive results across downstream tasks.

### 4.2. Compatibility of weights and image representations.

Beyond enabling alignment without paired data, we demonstrate that classification head weights are compatible with image representations, as these weights can be combined with real image-text pairs to augment the alignment dataset $\mathcal{D}_{imgtxt}$. In this setting, we define the augmented alignment dataset as

$$\mathcal{D}_{aug} = \mathcal{D}_{imgtxt} \bigcup \mathcal{D}_{weights} \qquad (3)$$

which is then used to learn $g$ and $\overline{g}$.

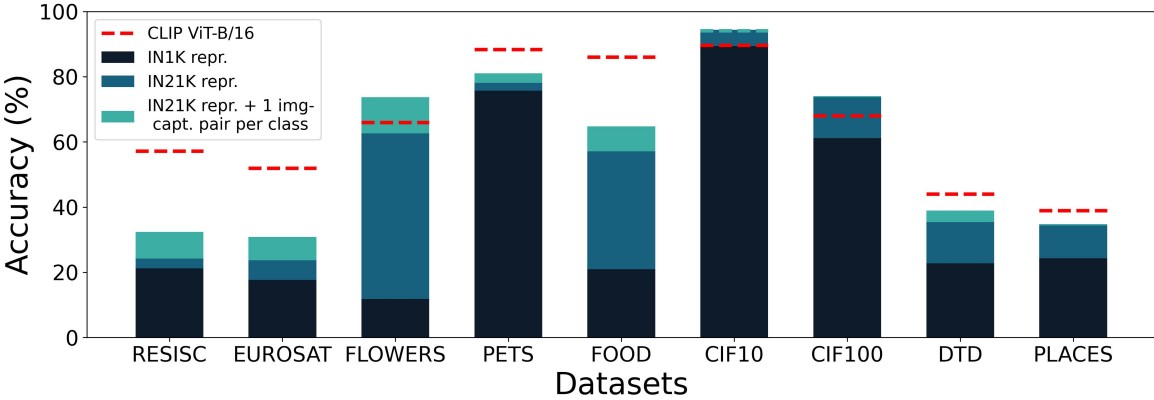

*Figure 3.* Zero-shot classification accuracy of BEiT-B/16 (Bao et al., 2021) aligned to text with an MLP. Stacked bars show the progressive accuracy gains: base performance when only classification head weights from ImageNet-1K pretraining are used during the MLP aligner training, additional gain from using ImageNet-21K weights, and further improvement from incorporating one image-caption pair per class, for all nine datasets at once. Red dashed lines show CLIP ViT-B/16 accuracy.

**Zero-shot classification.** In addition to the already discussed alignment without image-text pairs, Figure 3 also reports results when, following Equation (3) a small set of image-text pairs is used along the classification head weights from ImageNet-21K pretraining. Specifically, we add as $\mathcal{D}_{imgtxt}$ the pairs $(f_I(x), f_T(t))$, where one image $x$ is sampled from each class in each dataset and paired with an (LLM-generated) caption $t$. We note that we train a single MLP and then test zero-shot accuracy in all datasets.

Since the total sum of classes across all nine datasets is 817, the ratio between the size of $\mathcal{D}_{imgtxt}$ and the total alignment dataset $\mathcal{D}_{aug}$ is $\frac{817}{817+21,841} \approx 0.036$. Figure 3 shows that adding image-text representation pairs $\mathcal{D}_{imgtxt}$ further improves performance, but the gains vary by task dataset. Datasets with high class overlap with ImageNet-21K (e.g., CIFAR-10, CIFAR-100) show minimal improvements when including text-image pairs, suggesting these provide diminishing returns when visual categories are already well-represented in the weight representations $\mathcal{D}_{weights}$.

We note that the BEiT-B/16 zero-shot classifier is obtained with little additional data and computation once the pretrained encoders and classification head are available. With respect to the alignment step, augmenting with the weight representations from ImageNet-21K pretraining uses only 817 image-caption pairs, while the lightweight MLP requires only two GPU-minutes of training. Additionally, while ImageNet-21K pretraining weight representations may contain semantic overlap with downstream benchmarks, we note that CLIP has also likely seen most of the classes in the downstream classification benchmarks (Xu et al., 2024). For instance, (Xu et al., 2024) reconstructs CLIP's data curation process and finds over 700 out of the 1K classes in ImageNet-1K present in pretraining metadata and observes a correlation between downstream zero-shot classification accuracy and the number of classes matched the metadata.

**Cross-modal retrieval.** To further explore compatibility of image-text paired data and classification weight representations for post-hoc augmentation techniques, we conduct experiments on cross-modal retrieval. For the three alignment techniques (MLP, CSA and Text2Concepts), we use $\mathcal{D}_{aug}$ with $\mathcal{D}_{imgtxt}$ composed of FLICKR30K training samples (0 to 30 K pairs), then evaluate the aligned image and text encoders in the cross-modal retrieval task on the

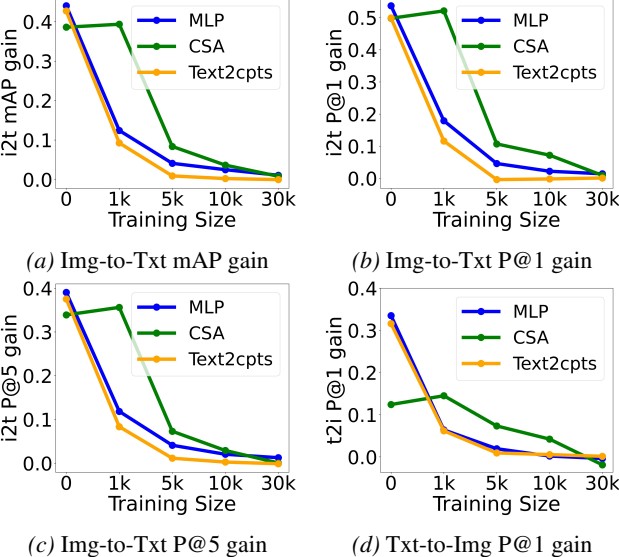

*(a)* Img-to-Txt mAP gain

*(b)* Img-to-Txt P@1 gain

*(c)* Img-to-Txt P@5 gain

*(d)* Txt-to-Img P@1 gain

*Figure 4.* Gains in FLICKR30K retrieval when augmenting image-text representations $D_{imgtxt}$ with classification head weights from ImageNet-21K pretraining $D_{weights}$. The x-axis shows the size of $D_{imgtxt}$. Post-hoc alignment techniques include CSA (Li et al., 2025), Text-to-Concept (Moayeri et al., 2023) and MLP alignment. Results show that augmenting with $\mathcal{D}_{weights}$ provides the largest gains in the low-data regime, with benefits diminishing as $\mathcal{D}_{imgtxt}$ grows—an expected outcome.

FLICKR30K test set. As more training image-text pairs from FLICKR30K are used in $\mathcal{D}_{imgtxt}$, the retrieval gains from augmenting the training set with classification head weights steadily diminish, tending towards zero when the full ~30K samples are used. This implies that augmenting the image-text pairs with the weight representations delivers the largest benefits in the low-data regime, with gains that diminish as more image-text pairs become available.

**Few-shot classification.** Beyond cross-modal retrieval and zero-shot classification, our approach can be used for few-shot classification through two main steps. (i) Initial alignment: Train a lightweight MLP (the same used in the retrieval and zero-shot setting) to map text representations to image representations using only the ImageNet-21K classification head (no image–text pairs). (ii) Fine-tuning on image–text data: Further fine-tune this MLP on $\mathcal{D}_{imgtxt}$, which consists of image–text pairs from the target few-shot dataset

$$\left(f_I(x),\ f_T(\text{``A photo of }\langle\text{class}\rangle\text{''})\right)$$

We employ sequential rather than joint training on the combined dataset $\mathcal{D}_{aug} = \mathcal{D}_{imgtxt} \bigcup \mathcal{D}_{weights}$ to make the alignment more focused on the specific classification domain of interest. Classification is done by finding the class such that the prompt *A photo of <class>* is most aligned with the image representation. We evaluate our approach against two strong baselines on frozen backbones: the Nearest Centroid Classifier (NCC) and K-Nearest Neighbors (KNN). The NCC, which computes class centroids from training representations and assigns test images to the nearest centroid, has been shown to be remarkably effective for few-shot classification, often surpassing more complex meta-learning approaches (Luo et al., 2023). We test across different classification tasks and standard few-shot settings with 1, 2 and 4 shots per class. Results in Table 2 (see experiments in additional datasets in Table 11 in the Appendix) show that our classifier performs significantly better than NCC and KNN in most cases. We acknowledge that a cosine classifier with simple text prompts (*A photo of <class>*) may be suboptimal for few-shot scenarios. Future work could explore prompt tuning or adapter-based approaches with the aligned vision-language model. However, our results demonstrate that even this straightforward approach surpasses strong baselines like NCC, validating the effectiveness of our weight recycling strategy for few-shot learning.

### 4.3. Further analyses.

In these additional analyses, we compare weights to image representations, evaluate how domain-specific weights affect downstream performance, and examine the geometric distribution of weight representations.

*Table 2.* Performance comparison across few-shot methods and number of shots per class $K$. Best results with statistical significance are in bold.

| K | Method | CIFAR10 | CIFAR100 | DTD | PLACES |
|---|--------|---------|----------|-----|--------|
| 1 | Ours | **93.34**±0.27 | **75.14**±0.25 | **47.36**±1.34 | **35.51**%±0.22 |
|   | NCC | 76.03±3.95 | 49.98±0.96 | 42.01±2.69 | 21.96%±0.71 |
|   | KNN | 76.03±3.95 | 49.98±0.96 | 42.01±2.69 | 21.96%±0.71 |
| 2 | Ours | **93.86**±0.49 | **75.77**±0.09 | 53.73±1.72 | **35.94**%±0.22 |
|   | NCC | 86.83±1.66 | 61.64±0.86 | 52.76±1.73 | 30.22%±0.51 |
|   | KNN | 71.67±4.65 | 48.50±0.80 | 41.58±3.04 | 22.49%±0.38 |
| 4 | Ours | **94.81**±0.36 | **76.42**±0.22 | 62.04±0.95 | **41.98**%±0.35 |
|   | NCC | 91.01±1.40 | 71.35±0.60 | 61.54±1.13 | 38.71%±0.36 |
|   | KNN | 84.69±1.15 | 62.41±0.56 | 51.64±1.51 | 28.77%±0.34 |

*Table 3.* Flickr30K zero-shot retrieval performance. We compare MLP-alignment recycling 1K ImageNet weight representations against the same number of averaged image representation pairs.

| Metric | Weight repr. | Image repr. |
|--------|-------------|-------------|
| i2t mAP | **0.2751** | 0.1860 |
| i2t P@1 | **0.3230** | 0.2110 |
| i2t P@5 | **0.2388** | 0.1566 |
| t2i P@1 | **0.2212** | 0.1406 |

**Weight representations outperform image representations under an equal alignment-data budget.** In Figure 1 we showed that classification head weights exhibit stronger semantic alignment with text embeddings than averaged image representations. This suggests that classification heads are superior anchors for vision-language alignment. To evaluate this claim in a downstream task, we compare the effectiveness of aligning using image-text pairs vs classification head weights. Specifically, we endow BEiT-B/16 with language capabilities via MLP-alignment using two different alignment datasets of equal size: (i) classification head weights from ImageNet-1K paired with their class names, and (ii) pairs composed of average image representations paired with their class name from ImageNet-1K. We then evaluate the aligned models on Flickr30K cross-modal retrieval. Results in Table 3 show that alignment using classification head weights consistently outperforms alignment using image-text pairs, despite the latter matching the downstream task's data modality. This confirms that recycled classification head weights are more effective for vision-language alignment than image representations, confirming in a downstream task our earlier analysis based on semantic alignment metric $m_{NN}$. Additional controls in the Appendix compare against one randomly sampled image representation per class and report a complementary zero-shot classification comparison (Tables 6 and 7).

**ImageNet-21K pretraining weights yield the best downstream performance** ImageNet-1K and ImageNet-21K are among the most widely employed publicly available labeled datasets for transfer learning (Ridnik et al., 2021).

While other datasets such as Places365 or iNaturalist have also been used for transfer learning (Plested & Gedeon, 2022), they are less general and provide far fewer off-the-shelf pretrained checkpoints compared to the ImageNet benchmarks. ImageNet-1K dataset offers a clean and balanced structure: all classes contain the same number of samples, and each class corresponds to a leaf synset in WordNet—specific nouns with no hyponyms in WordNet ontology (e.g., "chihuahua" but not "mammal")—ensuring semantic disjointness. By contrast, ImageNet-21K has an imbalanced class distribution and includes both leaf and non-leaf synsets (e.g., "chihuahua" and "mammal"), which introduces noise into the classification task from which the classification head weights are learned. Nonetheless, ImageNet-1K's clean, uniform structure comes at the cost of limited concept diversity. Instead, ImageNet-21K, with approximately 21 times more weights, provides a much richer set of semantic representations for vision-language alignment. This increased scale compensates for the possible noise caused by overlapping categories and class imbalance as seen in the better downstream performance shown in Figure 3. In addition to general domain datasets as ImageNet-1K and ImageNet-21K, we analyze how downstream performance is affected when using classification head weights from more specialized datasets. Section J in the Appendix presents experiments using classification head weights from iNaturalist-2021 (Van Horn et al., 2018), a large-scale dataset focused on natural species. While Table 12 in the Appendix shows recycling iNaturalist-2021 weights unlocks non-trivial vision-language alignment capabilities, performance on the classification benchmark is significantly lower than when using ImageNet-21K weights. This suggests that the generality and diversity of concepts in ImageNet-21K are key for effective vision-language alignment without paired data.

**Modality Gap Between Classification Head Weights and Image Representations.** Inspired by the text-image modality gap observed in models like CLIP (Liang et al., 2022), we investigate whether a similar separation exists between classification weights $w_i$s and image representations. Following the analysis procedure from the modality gap seminal work (Liang et al., 2022), our findings confirm a statistically significant gap: a permutation test on distances between centroids of both modalities showed that the observed distance (0.1723) was substantially larger than expected by chance (mean 0.0446, p<0.001). Furthermore, a simple single-layer MLP trained to distinguish between the two representation types achieved near-perfect test accuracy (99.75%). This systematic separation suggests that despite semantic alignment, classification weights and image features occupy distinct regions in the latent space. Figure 5 displays a UMAP (McInnes et al., 2018) visualization of weights and image representations where a separation be-

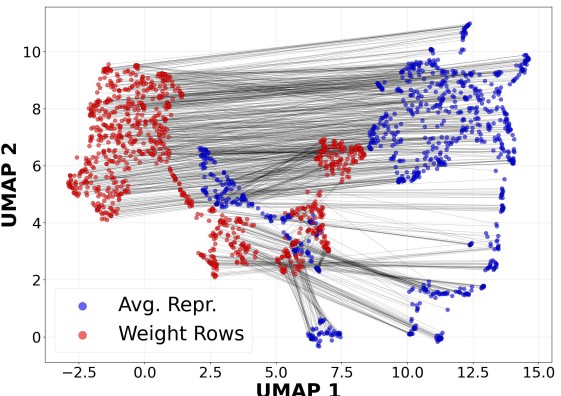

*Figure 5.* UMAP (McInnes et al., 2018) of class representations. Blue and red points show average image representation and weights $w_i$ respectively. Lines connect pairs from the same class.

tween the two modalities can be observed. Although we presented preliminary attempts to mitigate this gap in Section I (e.g., via centering and rescaling or lightweight projections), these simple strategies did not result in downstream tasks performance gains. We leave a more thorough investigation of this modality gap and its mitigation to future work. In spite of this gap, our results demonstrate that classification head weights are still compatible with image representations for vision-language alignment, as their combination boosts downstream tasks performance (e.g., Figure 6).

## 5. Limitations and Future Work

We aligned image and text encoders by combining image-text pairs and classification heads in the same dataset, as our goal was to demonstrate the compatibility of the two types of representations with this conceptually simple approach. However, we showed that an analogue to the well-known image-text modality gap (Liang et al., 2022) also occurs in the image representation space between images and row vectors $w_i$s of the classification head matrix. Although we presented preliminary attempts to mitigate this gap in Section I (e.g., via centering and rescaling or lightweight projections), these simple strategies did not result in performance gains. Thus, finding more sophisticated approaches for combining image-text pairs and classification weight representations remains an interesting future research direction to further improve alignment. Another natural extension would be to combine classifier heads from multiple supervised datasets. Such a combination is meaningful only when the heads are linear classifiers over the same frozen image feature space; otherwise, their row vectors live in incompatible coordinate systems. In addition, beyond the applications demonstrated in this work, a vision-language model enables capabilities that vision-only backbones cannot provide. For example, one could explore prompt tuning methodologies

such as CoOp (Zhou et al., 2022b) and CoCoOp (Zhou et al., 2022a), Visual QA (Song et al., 2022) or image captioning with text-only training (Li et al., 2023). Our approach opens the door to apply these methods to vision-only models in a fast, efficient way.

## 6. Conclusion

In this work, we have presented a novel framework for post-hoc vision-language alignment based on the observation that supervised classification head weights encode reusable semantic prototypes. By identifying and repurposing the classification head weights of pretrained vision models, we found a valuable source of representations that is typically discarded after pretraining. Our extensive experiments demonstrate that this weight recycling strategy is effective in two distinct regimes. First, it enables zero-shot alignment in the complete absence of image-text pairs during the alignment step, transforming unimodal vision backbones into vision-language models with minimal additional computation. Second, it serves as a powerful data augmentation technique that consistently boosts the performance of state-of-the-art post-hoc alignment methods in low-data scenarios, particularly for cross-modal retrieval and classification tasks. By reducing the additional data and compute barriers for post-hoc alignment, our approach contributes to the goals of Green AI. We hope this work inspires further research into the latent semantic potential of supervised pretraining weights and their application in resource-efficient representation learning.

## Acknowledgements

This work was supported by the Junta de Andalucía Predoctoral 2024 programme (DGP_PRED_2024_00569), under the Consejería de Universidad, Investigación e Innovación for predoctoral contracts for researchers in training, cofinanced by the Fondo Social Europeo Plus (FSE+) through the Andalucía 2021–2027 programme, the TSI-100927-2023-1 project under the Recovery, Transformation and Resilience Plan funded by the European Union NextGenerationEU through the Ministry for Digital Transformation and the Civil Service, and grants PID2023-149128NB-I00 and PID2023-150070NB-I00 funded by MICIU/AEI /10.13039/501100011033 and by ERDF, EU.

## Impact Statement

This work shows how classification heads, by-products of supervised pretraining, can be used to make vision-language alignment more data- and compute-efficient. This may broaden access to vision-language capabilities without requiring CLIP-scale paired-data training. However, the method inherits biases, label noise, licensing constraints, and semantic coverage limits from the pretrained models, classifier heads, and any image-text data used for alignment.

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

# A. Code

All code is available at: https://github.com/david-mnd/recycling4vlalignment

# B. Classification head weights row vectors as prototypes for the cosine classifier

In the main text, we provided empirical evidence (see Table 1) that the row vectors $w_i$ of a classification head $W$ can serve as effective class prototypes for a cosine classifier. We also briefly mentioned the Neural Collapse (NC) phenomenon (Papyan et al., 2020) as a potential theoretical basis, although the gaps between the ideal assumptions in NC and our setting lead us to provide experimental results as evidence. This appendix provides a more detailed discussion for using the classification weights $w_i$ as class prototypes in the cosine classifier, proceeding in two parts: first, an intuitive view, and second, a more formal justification via the properties of Neural Collapse.

**An intuitive view with degrees of freedom**. A linear classifier on a point $x$ is defined by $\operatorname{argmax}_i w_i^T x + b_i$. The decision is based on the *direction* of the weight vector $w_i$, its *magnitude* (norm) $||w_i||$, and the *bias* scalar $b_i$. On the other hand, in the cosine classifier the decision rule becomes

$$
\begin{aligned}
\operatorname{argmax}_i \cos(w_i, x) &= \operatorname{argmax}_i \left( \frac{w_i^T x}{||w_i||||x||} \right) \\
&= \operatorname{argmax}_i \left( \frac{w_i^T x}{||w_i||} \right)
\end{aligned}
\tag{4}
$$

In Equation (4), the norm and the bias take no part in the classification, as only the *directional* information of $w_i$ matters. For a $d$-dimensional feature space (e.g., $d = 768$), the score for each class depends on $d + 1$ parameters ($w_i \in \mathbb{R}^d$ and $b_i \in \mathbb{R}$). The normalized vector $w_i/||w_i||$ retains $d - 1$ degrees of freedom (the direction on the hypersphere), "losing" only 2 parameters (the magnitude and bias). The central hypothesis, supported by our empirical results in Table 1, is that the vast majority of the semantic, class-identifying information is encoded in the *direction* of this $d$-dimensional vector, not its magnitude or bias. The theoretical framework of Neural Collapse formalizes this intuition.

**Formal explanation via Neural Collapse (NC)**. The Neural Collapse (NC) phenomenon (Papyan et al., 2020) describes the geometric structure of the final-layer features and classifier weights during the terminal phase of cross-entropy training (i.e., after achieving near-zero training error). The key properties for us are[1]:

- *(NC1) Variability Collapse:* The feature vectors $x$ for all training samples belonging to a class $i$ collapse to their class mean, or centroid, $\mu_i$.

- *(NC3) Convergence to Self-Duality:* The classifier's weight vectors $w_i$ (from the $Wx + b$ layer) converge to be their corresponding class centroids $\mu_i$ up to rescaling by a scalar.

- *(NC4) Simplification to Nearest Class-Center:* The $Wx + b$ classifier converges to be equivalent to a nearest-centroid classifier, i.e., $\operatorname{argmax}_i(w_i^T x + b_i) \to \operatorname{argmin}_i ||x - \mu_i||$

We show these conditions lead to an equivalence between the linear classifier and the cosine classifier using $w_i$s rows as representations. First, we assume as in our setting, our features $x$ to be L2-normalized, so $||x|| = 1$. From *(NC1)*, all features in a class $i$ collapse to their centroid $\mu_i$, which implies in the training limit the centroids themselves must also be normalized (as they are equal to any of the representations of the points in the class), so $||\mu_i|| = 1$. From *(NC3)*, the weights $w_i$ align with these centroids $\mu_i$; given that $||\mu_i|| = 1$, the "up to rescaling" property simplifies, and the *normalized weight vector* and the class *centroid* converge to each other: $\mu_i \to w_i/||w_i||$. Finally, from *(NC4)*, the $Wx + b$ classifier converges to a nearest-centroid classifier, $\operatorname{argmax}_i(w_i^T x + b_i) \to \operatorname{argmin}_i ||x - \mu_i||^2$. By substituting the result from *(NC3)* into *(NC4)*, we see that the $Wx + b$ classifier, at convergence, becomes equivalent to a classifier that finds the nearest *normalized weight vector*: $\operatorname{argmax}_i(w_i^T x + b_i) \to \operatorname{argmin}_i ||x - w_i/||w_i||\,||$. Now we see this is the same as the cosine classifier on the rows of

---

[1]NC2 refers to the geometric configuration of class centroids and classifier's weight vectors, which is not relevant for our purposes.

the weight matrix. Since we assumed $||x|| = 1$, we have:

$$
\begin{aligned}
\left\| x - \frac{w_i}{||w_i||} \right\|^2 &= ||x||^2 - 2\frac{w_i^T x}{||w_i||} + \left\| \frac{w_i}{||w_i||} \right\|^2 \\
&= 1 - 2\frac{w_i^T x}{||w_i||} + 1 \\
&= 2(1 - \frac{w_i^T x}{||w_i||})
\end{aligned}
\tag{5}
$$

Hence

$$
\begin{aligned}
\operatorname{argmin}_i \left\| x - \frac{w_i}{||w_i||} \right\| &= \operatorname{argmin}_i \left\| x - \frac{w_i}{||w_i||} \right\|^2 \\
&\overset{(5)}{=} \operatorname{argmin}_i \left( 2(1 - \frac{w_i^T x}{||w_i||}) \right) \\
&= \operatorname{argmax}_i \left( \frac{w_i^T x}{||w_i||} \right) \\
&\overset{(4)}{=} \operatorname{argmax}_i \cos\left( w_i, x \right)
\end{aligned}
\tag{6}
$$

Thus $\operatorname{argmax}_i(w_i^T x + b_i) \rightarrow \operatorname{argmax}_i \cos(w_i, x)$, that is, the linear classifier becomes the cosine classifier on the rows of the weight matrix. As noted in the main text, the formal theory of Neural Collapse relies on strong assumptions (e.g., balanced data, number of classes less than the feature dimension, zero training error). While recent extensions address some of these limitations individually by adding new technical conditions (Jiang et al., 2024; Yang et al., 2022), to the best of our knowledge, there are still gaps between theoretical assumptions and our setting. This is why the direct empirical validation in Table 1 remains a crucial piece of evidence supporting our approach.

## C. Datasets

**Retrieval.** Evaluation of our data augmentation technique in retrieval setting is carried out on Flickr30K (Plummer et al., 2015), a widely-used benchmark for image-text retrieval tasks. The dataset contains 31,000 images collected from Flickr, each annotated with five human-written captions describing the visual content. To further validate the effectiveness of our approach, we also provide extended retrieval experiments on the MS-COCO dataset (Lin et al., 2014) in the appendix. MS-COCO is a large-scale dataset comprising over 120,000 images with diverse scenes and objects, each paired with five human-annotated captions. For both datasets, we follow the standard evaluation protocol using the widely-employed Karpathy splits (Karpathy & Fei-Fei, 2015).

**Classification.** We evaluate our approach for zero- and few-shot classification tasks in nine diverse datasets spanning multiple domains including remote sensing, natural scenes, objects, animals, food, and textures. All datasets are used with their standard splits where available. We utilize the `torchvision` implementation for all datasets except RESISC45, which is obtained from the `torchgeo` library (Stewart et al., 2024).

**RESISC45** (Cheng et al., 2017) is a remote sensing image scene classification dataset containing 31,500 images across 45 scene classes, with 700 images per class at 256×256 pixel resolution.

**EuroSAT** (Helber et al., 2019) consists of 27,000 Sentinel-2 satellite images covering 10 land use and land cover classes across European cities, with each image having 13 spectral bands at 64×64 pixels.

**Flowers102** (Nilsback & Zisserman, 2008) contains 8,189 images of 102 flower categories commonly found in the United Kingdom, with high intra-class variation and inter-class similarity.

**OxfordPets** (Parkhi et al., 2012) includes 7,349 images of 37 cat and dog breeds, providing a fine-grained classification challenge for animal recognition.

**Food101** (Bossard et al., 2014) comprises 101,000 images across 101 food categories, with 1,000 images per class, representing popular dishes from around the world.

**CIFAR-10 and CIFAR-100** (Krizhevsky & Hinton, 2009) are widely-used benchmark datasets containing 60,000 32×32 color images. CIFAR-10 has 10 classes while CIFAR-100 has 100 classes, both with 6,000 and 600 images per class respectively.

**DTD** (Cimpoi et al., 2014) (Describable Textures Dataset) contains 5,640 texture images organized into 47 categories, focusing on describable texture attributes rather than materials.

**Places365** (Zhou et al., 2018) is a scene-centric database designed for scene recognition and understanding. We use the Places365 validation set (50 images per class) instead of the larger test set (900 images per class) for computational efficiency. For zero-shot experiments, we evaluate on all 50 images per class. For few-shot experiments, we split each class into 10 training and 40 test images, then sample $N \in \{1, 2, 4, 8\}$ shots per class from the training set and evaluate on the 40 test images in each class.

## D. Metrics

In this work, we evaluated performance on both retrieval and classification tasks. Below we define each metric.

**Retrieval.** For image–text and text–image retrieval, we employ the same standard metrics as in related work (Li et al., 2025).

Mean Average Precision (mAP) Mean Average Precision summarizes the area under the precision–recall curve across all queries. For a single query $q$,

$$\text{AP}(q) = \frac{1}{R_q} \sum_{k=1}^{N} P(k)\, \mathbf{1}\{\text{relevant at rank } k\},$$

where $R_q$ is the number of relevant items for $q$, $P(k)$ is precision at cutoff $k$, $N$ is the length of the retrieved list and $\mathbf{1}\{\cdot\}$ is the indicator function that equals 1 if the item at rank $k$ is relevant, and 0 otherwise. The mAP over $Q$ queries is then

$$\text{mAP} = \frac{1}{Q} \sum_{q=1}^{Q} \text{AP}(q).$$

Note: In the Flickr30K dataset each image has 5 associated captions, so $R_q = 5$ for each image query.

Precision at $K$ (P@K) Precision at rank $K$ measures the proportion of relevant items among the top-$K$ results:

$$P@K = \frac{\text{Number of relevant items in top } K}{K}.$$

We report:

- *Image→Text*: $P@1$ and $P@5$

- *Text→Image*: $P@1$

While we compute these retrieval metrics under both baseline and augmented conditions, in the main article we present metric gains (e.g. increase in mAP, and in P@1/5). This succinctly highlights the impact of our augmentation strategy.

**Classification accuracy** For our classification experiments, we report *Accuracy*, defined as

$$\text{Accuracy} = 100 \times \frac{\text{Number of correctly classified samples}}{\text{Total number of samples}}$$

Accuracy is the most intuitive and widely used metric for balanced multi-class settings.

## E. Post-hoc Vision-Language Alignment

Formally, given an image encoder $f_I$ and a text encoder $f_T$, we learn alignment functions $g$ and $\bar{g}$ such that $g \circ f_I$ and $\bar{g} \circ f_T$ map images and texts into a shared latent space. In this space, images and texts with the same semantic content are assigned

similar embeddings, while those with different semantic concepts are placed far apart. These functions $g$ and $\bar{g}$ are learned using various methods, including CSA, text-to-concept, and MLP alignment.

To centralize the information on the post-hoc alignment methods employed in this work, we summarize here how they are used with recycled classifier weights. In the weight-only setting, we form the alignment dataset $\mathcal{D}_{weights} = \{(w_i, f_T(t_i))\}_{i=1}^{C}$, where $w_i$ is the classifier-weight vector associated with class $i$ and $t_i$ is the corresponding class name. For CSA, these pairs are used as standard paired samples and Canonical Correlation Analysis learns two linear mappings $g$ and $\bar{g}$ into a shared latent space. For text-to-concepts, we use the standard variant that learns a single linear mapping from text representations into the image/weight representation space. For MLP-alignment, we analogously learn a two-layer MLP from text representations into the image/weight representation space, leaving the image side unchanged. In the augmented setting, the same method is applied after replacing $\mathcal{D}_{weights}$ by $\mathcal{D}_{aug} = \mathcal{D}_{imgtxt} \cup \mathcal{D}_{weights}$.

We provide some further information on the hyperparameter choices for the different alignment methodologies. MLP-alignment consists of learning a lightweight MLP to map text to image representations. The lightweight MLP consists of a two-layer multilayer perceptron with GELU (Hendrycks & Gimpel, 2023) activation that maps text representations into the image feature space. The hidden dimension is 4 times that of the input dimension. The MLP is trained using the AdamW (Loshchilov & Hutter, 2019) optimizer with an initial learning rate of $5 \cdot 10^{-3}$ and cosine loss. We found that performance for retrieval and classification downstream tasks plateaued after around 500 epochs, with little change when training for longer, so we fixed the number of epochs to 500. We anneal the learning rate following a cosine schedule without restarts (Loshchilov & Hutter, 2017). Regarding the hyperparameters of CSA and text-to-concepts methods, the CSA method (Li et al., 2025) is implemented with the dimension of the shared representation space equal to 200 as we found out this was where downstream performance in cross-modal retrieval peaked. Similarly, the text-to-concept method is trained with MSE loss using an AdamW with cosine annealing schedule for 500 epochs. Since only lightweight MLPs are trained (two layers in the case of MLP-alignment and a single layer for text-to-concepts), the required training computational power is minimal. All experiments have been conducted on a GTX Titan Xp 12GB GPU where these lightweight models take at most a couple of minutes to train. For CSA, the heaviest operation is an SVD computation, which does not require GPU. All vision models are obtained from the checkpoint associated to ImageNet-21K task in `Timm` (Wightman, 2019) library. The checkpoint for the text encoder of CLIP ViT-B/32 model is that from `torch.hub`.

## F. Cross-modal retrieval

Cross-modal retrieval operates on a set of candidate texts and images within a shared latent space. Given a query, the system retrieves the target whose embedding exhibits the highest similarity to the query. In text-to-image retrieval, the system selects the image with the highest similarity to the query text embedding; image-to-text retrieval follows the inverse procedure. The cross-modal retrieval results are reported for the two settings considered in the article: (i) vision-language encoders aligned solely through weight representations (where $g$ and $\bar{g}$ are learned without image-text pairs); and (ii) alignment augmented by weight representations, where we analyze the improvements gained from weight recycling. Table 4 shows the results for the first setting, where the BEiT-B/16 image encoder is endowed with vision-language capabilities by aligning it with CLIP's text encoder. We observe that post-hoc alignment based on classification head representations yields non-trivial cross-modal retrieval performance. This demonstrates that semantic structures emerging during large-scale supervised pretraining can be leveraged to adapt the image backbone for these tasks without any paired data. Conversely, Table 5 presents the results for the second setting (specifically, the absolute values corresponding to the gains illustrated in Figure 4). Results for cross-modal retrieval when aligning other image encoders are shown in Table 5 (BEiT-B/16), Table 13 (CAFormer-S18), Table 14 (ConvFormer-S18), Table 15 (ConvNeXt-Base), and Table 16 (TinyViT-21M).

**Weight representations versus randomly sampled image representations.**    To complement the main-text analysis in Table 3, we address the concern that averaging image representations could remove instance-specific information useful for retrieval. We therefore compare MLP-alignment trained with ImageNet-1K classifier weights against the same protocol trained with one randomly sampled ImageNet-1K image representation per class, paired with the corresponding class name. We repeat the random-image sampling over multiple seeds and report mean and standard deviation. As shown in Table 6, classifier weights provide substantially stronger alignment data for Flickr30K retrieval than an equal number of sampled image representations.

*Table 4.* Zero-shot cross-modal retrieval for different alignment methods using only classification weights representations.

| Metric | Alignment Method | Datasets Flickr30K | COCO |
|---|---|---|---|
| i2t mAP | CSA | 0.3860 | 0.1842 |
| | MLP | 0.4404 | 0.2075 |
| | Text2cpts | 0.4271 | 0.2092 |
| | Random | 0.0026 | 0.0006 |
| i2t P@1 | CSA | 0.4980 | 0.2442 |
| | MLP | 0.5370 | 0.2562 |
| | Text2cpts | 0.4990 | 0.2540 |
| | Random | 0.0010 | 0.0002 |
| i2t P@5 | CSA | 0.3390 | 0.1668 |
| | MLP | 0.3904 | 0.1854 |
| | Text2cpts | 0.3754 | 0.1846 |
| | Random | 0.0010 | 0.0002 |
| t2i P@1 | CSA | 0.1242 | 0.0570 |
| | MLP | 0.3350 | 0.1556 |
| | Text2cpts | 0.3160 | 0.1433 |
| | Random | 0.0010 | 0.0002 |

*Table 5.* Cross-modal retrieval in Flickr30K for BEiT-B/16 aligned to CLIP-ViT-B/32's text encoder for different alignment methods (with and without weight augmentation) and varying training sample sizes. Metrics evaluated are: Image to text Mean Average Precision (mAP), Top-1 Precision (P@1) and Top-5 Precision (P@5) and Text to Image Top-1 Precision (P@1).

| Img. encoder | Metric | Alignment method | +ImgNet 21K | Training set size 0 | 1K | 5K | 10K | 30K |
|---|---|---|---|---|---|---|---|---|
| **BEiT-B/16** | i2t mAP | CSA | No | | 0.0668 | 0.3991 | 0.4561 | 0.4985 |
| | | | Yes | **0.3860** | **0.4603** | **0.4829** | **0.4926** | **0.5067** |
| | | MLP | No | | 0.3702 | 0.5090 | 0.5481 | 0.5903 |
| | | | Yes | **0.4404** | **0.4945** | **0.5500** | **0.5730** | **0.6011** |
| | | Text2cpts | No | | 0.4316 | 0.5563 | 0.5758 | **0.5895** |
| | | | Yes | **0.4271** | **0.5243** | **0.5654** | **0.5782** | 0.5893 |
| | i2t P@1 | CSA | No | | 0.0690 | 0.5110 | 0.5780 | 0.6410 |
| | | | Yes | **0.4980** | **0.5900** | **0.6180** | **0.6500** | **0.6500** |
| | | MLP | No | | 0.4450 | 0.6260 | 0.6730 | 0.7160 |
| | | | Yes | **0.5370** | 0.6240 | 0.6720 | **0.6950** | **0.7300** |
| | | Text2cpts | No | | 0.5270 | **0.6910** | **0.7100** | 0.7370 |
| | | | Yes | **0.4990** | **0.6430** | 0.6870 | 0.7080 | **0.7380** |
| | i2t P@5 | CSA | No | | 0.0558 | 0.3560 | 0.4074 | 0.4514 |
| | | | Yes | **0.3390** | **0.4120** | **0.4290** | **0.4364** | **0.4520** |
| | | MLP | No | | 0.3246 | 0.4534 | 0.4920 | 0.5260 |
| | | | Yes | **0.3904** | **0.4430** | **0.4944** | **0.5124** | **0.5386** |
| | | Text2cpts | No | | 0.3856 | 0.4976 | 0.5180 | **0.5252** |
| | | | Yes | **0.3754** | **0.4690** | **0.5090** | **0.5206** | 0.5240 |
| | t2i P@1 | CSA | No | | 0.0038 | 0.1070 | 0.1428 | **0.2432** |
| | | | Yes | **0.1242** | **0.1488** | **0.1804** | **0.1850** | 0.2244 |
| | | MLP | No | | 0.2766 | 0.3548 | 0.3766 | **0.3964** |
| | | | Yes | **0.3350** | **0.3406** | **0.3742** | **0.3786** | 0.3926 |
| | | Text2cpts | No | | 0.2786 | 0.3486 | 0.3574 | 0.3622 |
| | | | Yes | **0.3160** | **0.3404** | **0.3580** | **0.3628** | **0.3636** |

*Table 6.* Flickr30K zero-shot retrieval performance for MLP-alignment trained on 1K ImageNet classifier weight representations versus one randomly sampled ImageNet image representation per class.

| Metric | Weight repr. | Image repr. |
|--------|--------------|-------------|
| i2t mAP | **0.2751** | $0.1311 \pm 0.0074$ |
| i2t P@1 | **0.3230** | $0.1486 \pm 0.0152$ |
| i2t P@5 | **0.2388** | $0.1126 \pm 0.0065$ |
| t2i P@1 | **0.2212** | $0.0942 \pm 0.0038$ |

*Table 7.* Zero-shot classification accuracy (%) for MLP-alignment trained on ImageNet-1K classifier weight representations versus one randomly sampled ImageNet image representation per class.

| Dataset | Weight repr. | Image repr. |
|---------|--------------|-------------|
| RESISC | **21.16** | $14.19 \pm 2.97$ |
| EuroSAT | **17.70** | $16.18 \pm 2.58$ |
| Flowers | **11.82** | $5.90 \pm 2.38$ |
| Pets | **75.69** | $74.50 \pm 1.48$ |
| Food | **20.96** | $12.92 \pm 1.14$ |
| CIFAR10 | **89.35** | $87.64 \pm 1.17$ |
| CIFAR100 | **61.12** | $52.78 \pm 1.13$ |
| DTD | **22.71** | $15.43 \pm 1.25$ |
| Places | **24.32** | $15.24 \pm 0.91$ |

# G. Zero-shot classification

This experiment expands the zero-shot classification evaluation from the Experiments section in the paper by assessing aligned models zero-shot performance across a broader range of text encoders. The training details are exactly the same as those explained in the cross-modal retrieval (Section F).

As usual, we consider two alignment settings: (i) image and text encoders aligned solely through weight representations (where $g$ and $\bar{g}$ are learned without image-text pairs); and (ii) alignment augmented by weight representations, where we analyze the improvements gained from weight recycling. In the first setting, we conduct experiments across the nine classification benchmarks introduced in Section C. Five image encoders are evaluated (BEiT-B/16 (Bao et al., 2021), CAFormer-S18, ConvFormer-S18, ConvNeXt-B, and TinyViT-21M) when paired with four text encoders, CLIP ViT-B/32 text encoder, RoBERTa (all-roberta-large-v1), MPNet (all-mpnet-base-v2), and MiniLM (all-MiniLM-L6-v2), with the latter three from the `sentence-transformers` library. We also evaluated ResNet and ViT CLIP vision-language models (Radford et al., 2021), which were trained on >400M image-text pairs to achieve aligned representations.

For zero-shot evaluation, we follow the standard prompt-based classification procedure implemented in the released code. For each class, we encode the corresponding class prompt with the text encoder, map it with the learned alignment function, and assign a test image to the class whose aligned text representation has the highest cosine similarity with the aligned image representation.

**Weight representations versus image representations in zero-shot classification.** As a complementary check to the retrieval control in Table 6, we repeat the comparison on zero-shot classification benchmarks using the same MLP-alignment protocol. Table 7 shows that classifier weights also outperform randomly sampled image representations across all evaluated datasets.

Results presented in Table 10 show zero-shot accuracy when only ImageNet-21K pretraining classification weights are used to train the aligner MLP. The findings reveal consistent patterns across image encoders: the CLIP ViT-B/32 text encoder substantially outperforms text-only trained models, achieving best results in almost all configurations with performance gaps often exceeding 20-40 percentage points. Within each image encoder family, MPNet typically ranks second, followed by RoBERTa, while MiniLM generally shows the weakest performance. This superiority of accuracy could indicate CLIP's text encoder has learned to represent language in a way that's inherently connected to visual concepts through its training on image-caption pairs. The visual semantic understanding remains transferable even when the CLIP's text encoder is paired with completely different image encoders from the one it was originally trained with. This suggests that CLIP's

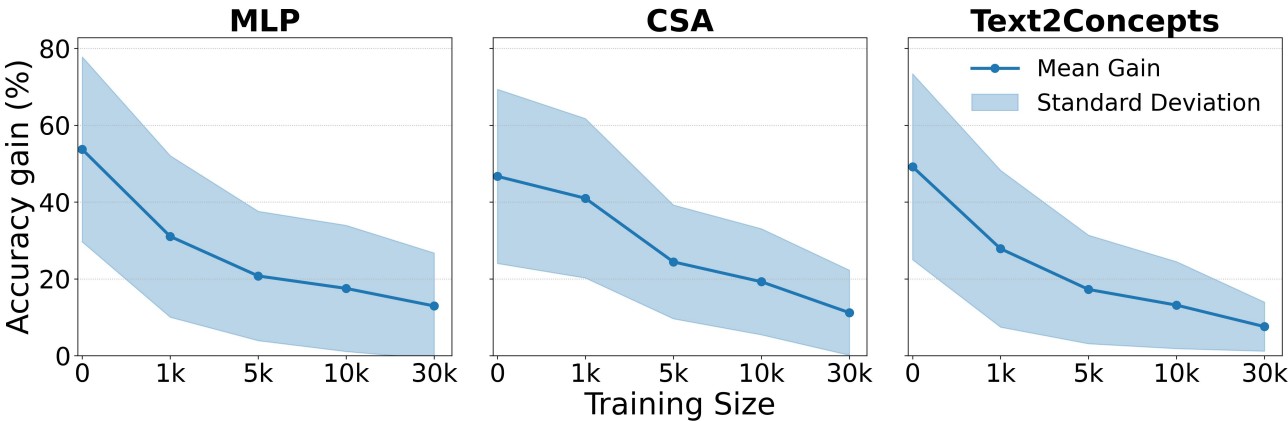

*Figure 6.* Mean zero-shot accuracy gains for BEiT-B/16 image encoder aligned to text on Flickr30K image-text pairs vs when augmenting alignment data with recycled ImageNet-21K weight representations. Zero-shot accuracy evaluated on: RESISC45, EuroSAT, Flowers102, OxfordPets, Food101, CIFAR-10, CIFAR-100, DTD, and Places365. The shaded region indicates the standard deviation across datasets.

advantage may not stem solely from jointly optimizing vision and text encoders, but specifically from CLIP's text encoder having learned to represent visual concepts in a much more effective way. In contrast, sentence-transformer models, despite excelling at capturing semantic relationships in text, lack this visual grounding, since they were trained purely on textual data without any visual context.

Regarding the setting in which weight representations are used together with image-text pairs for alignment, Table 8 presents the data from Figure 3. As an additional experiment, use Flickr30K image-text pairs to align the BEiT-B/16 image encoder to text, and compare zero-shot classification accuracy when using ImageNet-21K weights to augment this alignment data. The results, illustrated in Figure 6, demonstrate that incorporating weight representations alongside image-text pairs consistently enhances zero-shot classification accuracy across all post-hoc alignment methods. This further confirms compatibility between weight representations and image-text pairs for vision-language alignment.

**Zero-shot classification accuracy in a specialized domain dataset**. In the setting of post-hoc alignment using only weight representations from ImageNet-21K pretraining, we evaluate downstream zero-shot classification on a specialized domain dataset that exhibits a significant distribution shift from ImageNet-21K. Specifically, we employ the HAM10000 dataset (Tschandl et al., 2018), a benchmark for dermatological skin lesion image classification comprising 7 classes. Table 9 compares our method (BEiT-B/16 aligned via recycled ImageNet-21K weights) against ViT CLIP baselines and a random classification baseline. We report Balanced Accuracy to account for class imbalance. The results indicate that the specialized nature of dermatological images presents a challenge for all evaluated Vision-Language Models. Both CLIP and our proposed method achieve performance only marginally above the random baseline (14.28%). In particular, our method achieves a balanced accuracy of 18.32%, similar to the CLIP ViT-B/32 variant (18.85%). This suggests that the alignment learned from ImageNet-21K weights is as robust as CLIP's text-image alignment when applied to unseen images in specialized domains. In addition, these results confirm that for such specific tasks, fine-tuning is necessary for any general-purpose VLM.

*Table 8.* Zero-shot classification accuracy. BEiT-B/16 encoders are aligned to text via MLP training using different representations: ImageNet-1K classification head weights, ImageNet-21K weights, and ImageNet-21K weights enhanced with one image-caption pair per class. CLIP ViT-B/16 results are included as a natively text-aligned reference model (Radford et al., 2021). Bold indicates best performance per dataset; underlined indicates second-best.

| Image Encoder | Alignment train data | RESISC | EUROSAT | FLOWERS | PETS | FOOD | CIFAR10 | CIFAR100 | DTD | PLACES |
|---|---|---|---|---|---|---|---|---|---|---|
| | IN1K repr. | 21.16 | 17.70 | 11.82 | 75.69 | 20.96 | 89.35 | 61.12 | 22.71 | 24.32 |
| BEiT-B/16 | IN21K repr. | 24.21 | 23.67 | 62.63 | 78.06 | 57.12 | **94.40** | 73.66 | 35.37 | 34.34 |
| | IN21K repr. + 1 img-capt. pair per class | 32.37 | 30.85 | **73.74** | 80.98 | 64.76 | 94.11 | **74.04** | 38.94 | 34.78 |
| CLIP ViT-B/16 | WIT400M | **57.14** | **51.93** | 65.96 | **88.31** | **86.04** | 89.67 | 68.05 | **43.99** | **38.96** |

*Table 9.* Zero-shot Balanced Accuracy on the HAM10000 (Tschandl et al., 2018) dataset. We compare post-hoc alignment on BEiT-B/16 with CLIP variants. We observe that all models struggle with this specialized domain, yet our weight-recycling approach performs on par with the CLIP variants.

| Model | Balanced Accuracy (%) |
|---|---|
| Random | 14.28 |
| CLIP ViT-B/32 | **18.85** |
| CLIP ViT-L/14@336px | 17.72 |
| BEiT-B/16 (Ours) | 18.32 |

*Table 10.* Zero-shot classification accuracy (%) of image and text encoders aligned with an MLP that was trained only on classification heads of ImageNet-21K pretraining. Checkpoint of text models: CLIP = Text encoder of CLIP ViT-B/32 model in `torch.hub`, RoBERTa = all-roberta-large-v1, MPNet = all-mpnet-base-v2, MiniLM = all-MiniLM-L6-v2. Best result for each image encoder in each dataset is shown in **bold** and second-best is underlined. For reference, we evaluated ResNet and ViT CLIP vision-language models (Radford et al., 2021), which were trained on ≈400M image-text pairs to achieve aligned representations.

| Img. encoder | Txt. encoder | RESISC | EUROSAT | FLOWERS | PETS | FOOD | CIFAR10 | CIFAR100 | DTD | PLACES |
|---|---|---|---|---|---|---|---|---|---|---|
| BEiT-B/16 | CLIP | **24.21** | **23.67** | **62.63** | **78.06** | **57.12** | **94.40** | **73.66** | 35.37 | **34.34** |
| BEiT-B/16 | RoBERTa | 18.54 | 17.15 | 38.54 | 35.32 | 29.48 | 92.12 | 65.93 | 16.28 | 25.86 |
| BEiT-B/16 | MPNet | 20.95 | 18.15 | 35.75 | 36.14 | 27.20 | 92.70 | 67.37 | 18.78 | 26.29 |
| BEiT-B/16 | MiniLM | 14.35 | 12.85 | 31.86 | 28.48 | 25.24 | 92.29 | 63.58 | 15.85 | 20.61 |
| CAFormer-S18 | CLIP | 16.89 | **22.70** | **67.44** | **75.20** | **59.83** | 86.05 | **69.08** | 37.39 | **32.85** |
| CAFormer-S18 | RoBERTa | 16.11 | 20.70 | 40.80 | 33.96 | 34.54 | **89.33** | 66.15 | 12.87 | 27.33 |
| CAFormer-S18 | MPNet | **18.52** | 20.48 | 42.92 | 46.96 | 32.69 | 86.76 | 67.36 | 15.05 | 25.28 |
| CAFormer-S18 | MiniLM | 11.27 | 17.37 | 38.87 | 37.97 | 20.10 | 85.64 | 62.09 | 15.21 | 20.27 |
| ConvFormer-S18 | CLIP | **19.71** | 23.93 | **66.12** | **75.47** | **54.92** | 85.03 | **65.85** | 37.82 | **32.67** |
| ConvFormer-S18 | RoBERTa | 17.33 | 16.44 | 35.47 | 38.81 | 32.30 | **85.98** | 62.40 | 12.71 | 26.57 |
| ConvFormer-S18 | MPNet | 18.52 | **27.00** | 33.62 | 45.68 | 30.61 | 82.31 | 59.09 | 16.65 | 24.63 |
| ConvFormer-S18 | MiniLM | 12.73 | 18.59 | 30.25 | 38.18 | 17.81 | 83.05 | 57.32 | 12.87 | 19.29 |
| ConvNext-B | CLIP | **27.29** | **24.78** | **69.21** | **73.78** | **60.29** | **92.34** | **71.66** | 35.85 | **35.09** |
| ConvNext-B | RoBERTa | 22.79 | 19.15 | 29.81 | 25.48 | 28.75 | 90.99 | 67.04 | 16.91 | 27.96 |
| ConvNext-B | MPNet | 24.65 | 23.63 | 40.80 | 42.00 | 30.52 | 91.81 | 68.68 | 19.73 | 28.48 |
| ConvNext-B | MiniLM | 21.73 | 17.00 | 39.70 | 26.27 | 27.55 | 90.97 | 64.64 | 18.83 | 22.61 |
| TinyViT-21M | CLIP | **28.33** | **26.48** | **68.42** | **76.81** | **57.30** | **89.69** | **69.39** | 35.59 | **33.80** |
| TinyViT-21M | RoBERTa | 22.75 | 21.48 | 34.90 | 26.76 | 22.09 | 86.94 | 65.05 | 14.84 | 25.91 |
| TinyViT-21M | MPNet | 24.59 | 25.56 | 33.22 | 25.59 | 26.59 | 88.98 | 64.22 | 18.94 | 26.09 |
| TinyViT-21M | MiniLM | 18.68 | 24.93 | 32.41 | 27.56 | 20.52 | 87.46 | 59.94 | 15.37 | 20.89 |
| CLIP RN-101 | | 41.37 | 26.70 | 61.99 | 86.24 | 79.83 | 77.77 | 48.52 | 39.68 | 36.00 |
| CLIP RN-50 | | 41.25 | 28.00 | 61.12 | 85.15 | 75.99 | 69.81 | 40.68 | 39.73 | 36.96 |
| CLIP ViT-B/16 | | 57.14 | 51.93 | 65.96 | 88.31 | 86.04 | 89.67 | 68.05 | 43.99 | 38.96 |
| CLIP ViT-B/32 | | 55.19 | 34.44 | 63.64 | 87.76 | 80.53 | 89.08 | 64.14 | 29.79 | 38.71 |

# H. Few-shot classification

*Table 11.* Full performance comparison across few-shot methods and number of shots per class $K$ across all nine datasets. Experiments were conducted using five random seeds, and results are reported as mean $\pm$ standard deviation. Best results with statistical significance are in bold.

| K | Method | RESISC | EUROSAT | FLOWERS | PETS | FOOD | CIFAR10 | CIFAR100 | DTD | PLACES |
|---|---|---|---|---|---|---|---|---|---|---|
| | Ours | 43.51±1.28 | 56.65±4.06 | 95.59±0.65 | **84.20±0.88** | 67.53±0.59 | 93.34±0.27 | 75.14±0.25 | 47.36±1.34 | 35.51%±0.22 |
| 1 | NCC | 39.27±3.50 | 56.38±2.61 | 94.49±0.75 | 76.09±3.63 | 51.71±0.92 | 76.03±3.95 | 49.98±0.96 | 42.01±2.69 | 21.96%±0.71 |
| | KNN | 39.27±3.50 | 56.38±2.61 | 94.49±0.75 | 76.09±3.63 | 51.71±0.92 | 76.03±3.95 | 49.98±0.96 | 42.01±2.69 | 21.96%±0.71 |
| | Ours | 52.14±0.95 | 64.59±3.82 | 97.41±0.70 | **85.98±0.51** | 70.96±0.61 | 93.86±0.49 | 75.77±0.09 | 53.73±1.72 | **35.94%±0.22** |
| 2 | NCC | 53.49±3.46 | 66.36±2.63 | 97.64±0.73 | 82.89±2.16 | 63.47±0.84 | 86.83±1.66 | 61.64±0.86 | 52.76±1.73 | 30.22%±0.51 |
| | KNN | 38.77±2.89 | 52.17±3.08 | 93.69±0.35 | 73.99±0.98 | 50.07±1.82 | 71.67±4.65 | 48.50±0.80 | 41.58±3.04 | 22.49%±0.38 |
| | Ours | 58.54±0.74 | 75.66±2.41 | 98.80±0.20 | 89.60±0.77 | 74.41±0.61 | **94.81±0.36** | 76.42±0.22 | 62.04±0.95 | **41.98%±0.35** |
| 4 | NCC | **64.53±0.97** | 76.27±2.50 | 98.91±0.34 | 88.17±1.78 | 73.60±0.83 | 91.01±1.40 | 71.35±0.60 | 61.54±1.13 | 38.71%±0.36 |
| | KNN | 51.26±1.13 | 67.25±2.36 | 97.08±0.38 | 82.73±1.67 | 63.49±0.89 | 84.69±1.15 | 62.41±0.56 | 51.64±1.51 | 28.77%±0.34 |

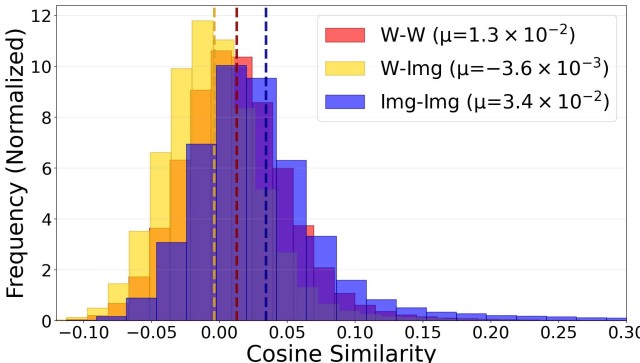

*Figure 7.* Distribution of the cosine similarities for averaged image representations and classification weights corresponding to the ImageNet1K classes. W-W indicates the distribution of cosine similarities between weights, Img-Img between average image representations, and W-Img inter-modality cosine similarities.

We evaluate our few-shot classification approach on the nine diverse classification tasks across standard shot settings with 1, 2, and 4 shots per class, comparing against Nearest Centroid Classifier (NCC) and K-Nearest Neighbors (KNN) baselines on frozen backbones. NCC has been shown to be remarkably effective for few-shot classification, often surpassing more complex meta-learning approaches (Luo et al., 2023). Our sequential training methodology employs the same lightweight MLP architecture as used in cross-modal retrieval and zero-shot classification settings, trained with identical optimizer, scheduler, and loss configurations from previous experiments. The initial alignment stage trains for 500 epochs using only ImageNet-21K classification weights, followed by 200 epochs of fine-tuning on target dataset image-text pairs. Once the MLP is trained, we predict the class of a given image $x$ by assigning it to the class text embedding is closest in cosine similarity to the MLP output for $x$. To assess statistical significance of accuracy improvements, we conduct five independent experimental runs where the exact same images are sampled for each class across all compared methods (ours, NCC, KNN) within each run, ensuring controlled comparison. We then perform paired t-tests on the resulting accuracy differences between methods, assuming normality of the accuracy distribution across runs, to validate the statistical significance of our method's gains over NCC and KNN. Results are shown in Table 11, where our method frequently outperforms both NCC and KNN baselines with statistical significance (p-value $< 0.05$). As noted in the main article, the classification method based on sequential training and cosine classifier is probably suboptimal. However, it serves to illustrate that even a naive approach can be competitive when combined with our weight augmentation method.

## I. Modality Gap Between Classification Head Weights and Image Representations

While vision-language models like CLIP demonstrate effective alignment between text and image representations in shared latent spaces, prior work has identified a modality gap where representations from image and text data occupy distinct regions (Liang et al., 2022). We investigate whether a similar phenomenon exists between row vectors $w_i$ of the classification head weight matrix $W$ and representations of images belonging to the corresponding classes. Specifically, we examine ImageNet-21K weights by extracting weight vectors for the 1K classes that correspond to ImageNet-1K, alongside averaged image representations computed from 50 images from each class in the ImageNet-1K. Following (Liang et al., 2022), we carry out two analyses: a permutation test of the centroid distances and an analysis of linear separability by training a single-layer MLP to distinguish between weight vectors and image representations. The permutation test evaluates whether the observed distance between modality centroids exceeds what would be expected by chance, by comparing the actual distance to a null distribution generated from random permutations of the data. Through this permutation testing of centroid distances, we observe a statistically significant separation: the distance between image and weight centroids (0.1723) substantially exceeds chance expectation (mean=0.0446, std=0.0014, p<0.001, Cohen's d=93.91). Additionally, by training a single-layer MLP classifier to discriminate between weight and image representations using an 80/20 train-test split, the classifier correctly classifies 199/200 test weight samples and 200/200 test image samples. These findings suggest that despite semantic alignment, there exists a systematic geometric separation between classification head row vectors and image features. As shown in Figure 7, the distribution of cosine similarities between weights and image representations (W-Img) is clearly shifted towards lower values compared to the distributions within each modality (W-W and Img-Img).

**Testing modality gap mitigation strategies**. Following the observation of the geometric separation between weight and

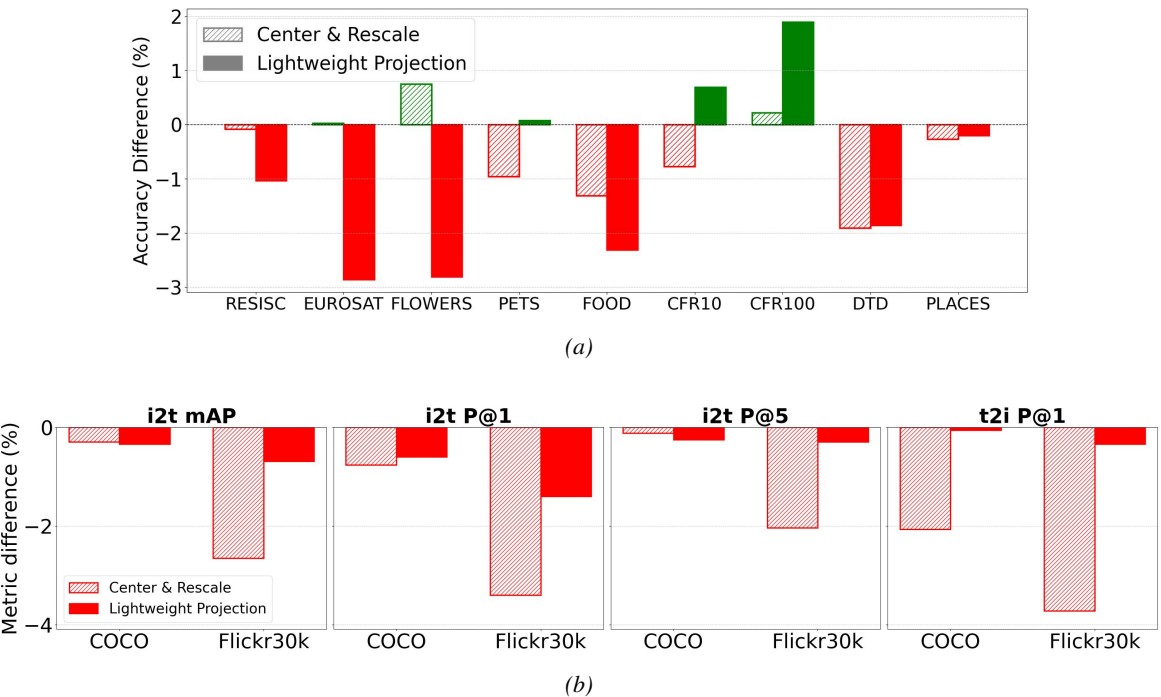

*Figure 8.* Change in zero-shot classification (a) and retrieval (b) performance when using basic modality gap mitigation strategies. ImageNet-1K weights representations undergoing two different modality-gap mitigation strategies (*Center & Rescale* and *Lightweight projection*) are used as alignment data. The results show that explicitly mitigating the geometric modality gap did not result in better downstream performance.

image representations, we conducted preliminary experiments to determine if bridging this *modality gap* prior to alignment enhances downstream performance. We compared the baseline strategy (union from Equation (3)) against two explicit mitigation approaches applied to the ImageNet-1K weight representations:

- Centering and rescaling: we subtract the mean of the weight representations (centering), adding the mean of the image representations (shifting towards the image modality) and rescaling the vectors to unit norm.

- Lightweight projection: we introduced lightweight linear projection layers designed to map weight representations onto the image manifold, trained alongside the alignment objective.

Figure 8 describes the increase in downstream zero-shot classification and retrieval performance across multiple benchmarks when modality-gap mitigation strategies are employed in alignment data. Our results indicate that the modality gap mitigation strategies do not improve downstream performance compared to the union approach from Equation (3). This is not surprising, as it aligns with observations in (Liang et al., 2022), which note that closing the modality gap does not necessarily guarantee better downstream performance. Consequently, since these mitigation techniques did not yield performance gains, we maintain the use of the original union strategy for this work and leave the exploration of more advanced gap mitigation techniques for future research.

## J. Alignment using classifier weights from a domain-specific dataset

In this section, we extend the evaluation of our proposed method by exploring how zero-shot classification accuracy varies when utilizing as alignment data weight representations from pretraining in domain-specific dataset iNaturalist (Van Horn et al., 2018). iNaturalist is a large-scale dataset characterized by fine-grained categories of flora and fauna, presenting a highly specialized semantic domain. The ConvNeXt checkpoint used as the image encoder was obtained from the `Timm` (Wightman, 2019) library.

As shown in Table 12, the downstream performance is notably higher when using ImageNet-21K. This superior performance is likely due to the greater semantic overlap between the diverse ImageNet-21K classes and the classes found in the

*Table 12.* A comparison of zero-shot classification accuracy (%) for iNaturalist and ImageNet-21K weights serving as alignment data for training. We use ConvNeXt as image encoder.

| Alignment method | Data | RESISC | EUROSAT | FLOWERS | PETS | FOOD | CIFAR10 | CIFAR100 | DTD | PLACES |
|---|---|---|---|---|---|---|---|---|---|---|
| CSA | ImageNet-21K | **20.60** | **18.93** | **52.20** | **60.70** | **50.05** | **90.56** | **65.46** | **27.71** | **30.72** |
| | iNaturalist | 4.68 | 16.33 | 37.62 | 20.85 | 5.28 | 30.07 | 14.46 | 6.76 | 1.11 |
| MLP | ImageNet-21K | **27.29** | **24.78** | **69.21** | **73.78** | **60.29** | **92.34** | **71.66** | **35.85** | **35.09** |
| | iNaturalist | 2.48 | 15.81 | 24.26 | 8.42 | 2.19 | 34.34 | 6.80 | 2.87 | 0.43 |
| Text2cpts | ImageNet-21K | **24.33** | **20.52** | **55.41** | **71.16** | **51.26** | **91.75** | **65.90** | **32.66** | **31.66** |
| | iNaturalist | 2.60 | 9.26 | 42.67 | 17.20 | 2.90 | 37.45 | 17.38 | 8.83 | 0.44 |

*Table 13.* Cross-modal retrieval in Flickr30K for CAFormer-S18 aligned to CLIP-ViT-B/32's text encoder for different alignment methods (with and without weight augmentation) and varying training sample sizes. Metrics evaluated are: Image to text Mean Average Precision (mAP), Top-1 Precision (P@1) and Top-5 Precision (P@5) and Text to Image Top-1 Precision (P@1).

| Img. encoder | Metric | Alignment method | +ImgNet 21K | Training set size | | | | |
|---|---|---|---|---|---|---|---|---|
| | | | | **0** | **1K** | **5K** | **10K** | **30K** |
| CAFormer-S18 | i2t mAP | CSA | No | | 0.0033 | 0.2936 | 0.3941 | 0.4582 |
| | | | Yes | 0.3182 | **0.4063** | **0.4275** | **0.4395** | **0.4635** |
| | | MLP | No | | 0.3197 | **0.4471** | 0.4950 | **0.5391** |
| | | | Yes | 0.3745 | 0.3833 | 0.4352 | **0.4963** | 0.5362 |
| | | Text2cpts | No | | 0.3974 | **0.5184** | **0.5360** | **0.5485** |
| | | | Yes | 0.3696 | 0.4727 | 0.5176 | 0.5311 | 0.5436 |
| | i2t P@1 | CSA | No | | 0.0020 | 0.3390 | 0.4970 | 0.5690 |
| | | | Yes | 0.4160 | 0.5020 | 0.5390 | 0.5440 | 0.5750 |
| | | MLP | No | | 0.3800 | 0.5460 | 0.6210 | 0.6730 |
| | | | Yes | 0.4440 | 0.4650 | 0.5260 | 0.6170 | 0.6600 |
| | | Text2cpts | No | | 0.4980 | 0.6400 | 0.6600 | **0.6780** |
| | | | Yes | 0.4700 | 0.6120 | 0.6470 | 0.6680 | 0.6700 |
| | i2t P@5 | CSA | No | | 0.0012 | 0.2640 | 0.3524 | 0.4120 |
| | | | Yes | 0.2818 | 0.3668 | 0.3816 | 0.3932 | 0.4168 |
| | | MLP | No | | 0.2770 | **0.3964** | 0.4404 | **0.4868** |
| | | | Yes | 0.3336 | 0.3400 | 0.3870 | **0.4418** | 0.4842 |
| | | Text2cpts | No | | 0.3496 | 0.4626 | **0.4796** | **0.4920** |
| | | | Yes | 0.3316 | 0.4268 | 0.4628 | 0.4754 | 0.4898 |
| | t2i P@1 | CSA | No | | 0.0018 | 0.0578 | 0.1616 | 0.2186 |
| | | | Yes | 0.0872 | 0.1260 | 0.1892 | 0.1772 | 0.2264 |
| | | MLP | No | | 0.2456 | 0.3474 | 0.3842 | 0.4124 |
| | | | Yes | 0.2892 | 0.2904 | 0.3752 | 0.3882 | 0.4126 |
| | | Text2cpts | No | | 0.2464 | **0.2946** | **0.3056** | **0.3140** |
| | | | Yes | 0.2160 | 0.2698 | 0.2892 | 0.2974 | 0.3098 |

downstream datasets. In contrast, the semantic variance within iNaturalist is limited to the biological domain. Consequently, it is reasonable to hypothesize that the alignment of the text and image encoders in the ImageNet-21K case is performed over a much broader and diverse region of the latent space. This hypothesis is supported by the observation that the iNaturalist-aligned models exhibit a clear domain preference: they achieve relatively higher accuracy on datasets containing plant and animal classes (such as Flowers102, Oxfordpets, and CIFAR (Nilsback & Zisserman, 2008; Parkhi et al., 2012; Krizhevsky & Hinton, 2009)) compared to non-biological datasets. However, it is important to note that despite this domain affinity, the absolute performance on these biological datasets remains lower than that of the ImageNet-21K-aligned models. Finally, we note that performance on some of the datasets lacking specific plant or animal content remains non-trivial, suggesting that despite the imperfect alignment, a meaningful amount of general semantic information is still being transferred.

*Table 14.* Cross-modal retrieval in Flickr30K for ConvFormer-S18 aligned to CLIP-ViT-B/32's text encoder for different alignment methods (with and without weight augmentation) and varying training sample sizes. Metrics evaluated are: Image to text Mean Average Precision (mAP), Top-1 Precision (P@1) and Top-5 Precision (P@5) and Text to Image Top-1 Precision (P@1).

| Img. encoder | Metric | Alignment method | +ImgNet 21K | Training set size | | | | |
|---|---|---|---|---|---|---|---|---|
| | | | | 0 | 1K | 5K | 10K | 30K |
| ConvFormer-S18 | i2t mAP | CSA | No | | 0.0027 | 0.2117 | 0.2933 | 0.3856 |
| | | | Yes | **0.3156** | **0.3979** | **0.4222** | **0.4396** | **0.4578** |
| | | MLP | No | | 0.3141 | **0.4468** | **0.4848** | **0.5321** |
| | | | Yes | **0.3956** | **0.3747** | 0.4392 | 0.4762 | 0.5302 |
| | | Text2cpts | No | | 0.3972 | 0.5145 | **0.5341** | **0.5475** |
| | | | Yes | **0.3807** | **0.4736** | 0.5174 | 0.5314 | 0.5436 |
| | i2t P@1 | CSA | No | | 0.0020 | 0.2510 | 0.3530 | 0.4840 |
| | | | Yes | **0.4130** | **0.5050** | **0.5520** | **0.5650** | **0.5680** |
| | | MLP | No | | 0.3820 | **0.5500** | 0.5840 | **0.6530** |
| | | | Yes | **0.4840** | 0.4660 | 0.5300 | **0.5940** | 0.6440 |
| | | Text2cpts | No | | 0.4940 | **0.6290** | **0.6520** | **0.6580** |
| | | | Yes | **0.4970** | **0.5940** | 0.6220 | 0.6470 | 0.6570 |
| | i2t P@5 | CSA | No | | 0.0006 | 0.1934 | 0.2646 | 0.3450 |
| | | | Yes | **0.2790** | **0.3544** | **0.3718** | **0.3912** | **0.4090** |
| | | MLP | No | | 0.2746 | **0.3928** | **0.4286** | **0.4796** |
| | | | Yes | **0.3498** | 0.3306 | 0.3912 | 0.4214 | 0.4786 |
| | | Text2cpts | No | | 0.3498 | 0.4582 | 0.4730 | **0.4894** |
| | | | Yes | **0.3396** | **0.4276** | 0.4670 | **0.4744** | 0.4884 |
| | t2i P@1 | CSA | No | | 0.0010 | 0.0652 | 0.0666 | 0.1424 |
| | | | Yes | **0.0976** | **0.1608** | **0.1642** | **0.2026** | **0.2144** |
| | | MLP | No | | 0.2432 | 0.3550 | 0.3834 | **0.4134** |
| | | | Yes | **0.3210** | 0.2794 | 0.3762 | 0.4100 | 0.4104 |
| | | Text2cpts | No | | 0.2376 | **0.2976** | **0.3140** | **0.3230** |
| | | | Yes | **0.2460** | **0.2724** | 0.2968 | 0.3098 | 0.3184 |

*Table 15.* Cross-modal retrieval in Flickr30K for ConvNeXt-Base aligned to CLIP-ViT-B/32's text encoder for different alignment methods (with and without weight augmentation) and varying training sample sizes. Metrics evaluated are: Image to text Mean Average Precision (mAP), Top-1 Precision (P@1) and Top-5 Precision (P@5) and Text to Image Top-1 Precision (P@1).

| Img. encoder | Metric | Alignment method | +ImgNet 21K | Training set size | | | | |
|---|---|---|---|---|---|---|---|---|
| | | | | 0 | 1K | 5K | 10K | 30K |
| ConvNeXt-Base | i2t mAP | CSA | No | | 0.0037 | 0.4125 | 0.4706 | 0.4972 |
| | | | Yes | **0.3723** | **0.4695** | **0.4948** | **0.5008** | **0.5065** |
| | | MLP | No | | 0.3722 | 0.5137 | 0.5475 | 0.5947 |
| | | | Yes | **0.4422** | **0.4962** | **0.5496** | **0.5772** | **0.6065** |
| | | Text2cpts | No | | 0.4580 | 0.5748 | 0.5912 | 0.6015 |
| | | | Yes | **0.4182** | **0.5333** | **0.5821** | **0.5946** | **0.6025** |
| | i2t P@1 | CSA | No | | 0.0020 | 0.5270 | 0.5730 | 0.6080 |
| | | | Yes | **0.4610** | **0.5860** | **0.6180** | **0.6250** | **0.6270** |
| | | MLP | No | | 0.4520 | 0.6300 | 0.6620 | 0.7350 |
| | | | Yes | **0.5240** | **0.6110** | **0.6800** | **0.7120** | **0.7520** |
| | | Text2cpts | No | | 0.5650 | **0.7130** | 0.7230 | 0.7230 |
| | | | Yes | **0.4710** | **0.6430** | 0.7090 | **0.7260** | **0.7290** |
| | i2t P@5 | CSA | No | | 0.0016 | 0.3656 | 0.4242 | 0.4412 |
| | | | Yes | **0.3352** | **0.4212** | **0.4440** | **0.4522** | **0.4554** |
| | | MLP | No | | 0.3296 | 0.4578 | 0.4892 | 0.5294 |
| | | | Yes | **0.3918** | **0.4434** | **0.4916** | **0.5134** | **0.5424** |
| | | Text2cpts | No | | 0.4060 | 0.5082 | 0.5278 | 0.5396 |
| | | | Yes | **0.3726** | **0.4824** | **0.5164** | **0.5346** | **0.5404** |
| | t2i P@1 | CSA | No | | 0.0014 | 0.1082 | 0.1892 | 0.2342 |
| | | | Yes | **0.1740** | **0.1968** | **0.2512** | **0.2668** | **0.2546** |
| | | MLP | No | | 0.2864 | 0.3796 | **0.4064** | **0.4350** |
| | | | Yes | **0.3594** | **0.3662** | **0.3948** | 0.4050 | 0.4306 |
| | | Text2cpts | No | | 0.2780 | **0.3408** | **0.3512** | **0.3624** |
| | | | Yes | **0.3114** | **0.3082** | 0.3368 | 0.3434 | 0.3592 |

*Table 16.* Cross-modal retrieval in Flickr30K for TinyViT-21M aligned to CLIP-ViT-B/32's text encoder for different alignment methods (with and without weight augmentation) and varying training sample sizes. Metrics evaluated are: Image to text Mean Average Precision (mAP), Top-1 Precision (P@1) and Top-5 Precision (P@5) and Text to Image Top-1 Precision (P@1).

| Img. encoder | Metric | Alignment method | +ImgNet 21K | Training set size | | | | |
|---|---|---|---|---|---|---|---|---|
| | | | | **0** | **1K** | **5K** | **10K** | **30K** |
| TinyViT-21M | i2t mAP | CSA | No | | 0.1501 | 0.4140 | 0.4578 | 0.4853 |
| | | | Yes | **0.3792** | **0.4551** | **0.4807** | **0.4885** | **0.4896** |
| | | MLP | No | | 0.3848 | 0.5032 | 0.5332 | 0.5721 |
| | | | Yes | **0.4375** | **0.4861** | **0.5293** | **0.5459** | **0.5751** |
| | | Text2cpts | No | | 0.4566 | 0.5450 | 0.5580 | 0.5678 |
| | | | Yes | **0.4389** | **0.5145** | **0.5534** | **0.5635** | **0.5709** |
| | i2t P@1 | CSA | No | | 0.1850 | 0.5230 | 0.5840 | **0.6280** |
| | | | Yes | **0.4780** | **0.5710** | **0.6040** | **0.6270** | 0.6200 |
| | | MLP | No | | 0.4780 | 0.6200 | **0.6740** | **0.6990** |
| | | | Yes | **0.5510** | **0.6210** | **0.6610** | 0.6610 | 0.6970 |
| | | Text2cpts | No | | 0.5900 | **0.6790** | 0.6970 | 0.7030 |
| | | | Yes | **0.5380** | **0.6260** | **0.6790** | **0.7010** | **0.7090** |
| | i2t P@5 | CSA | No | | 0.1314 | 0.3714 | 0.4082 | 0.4338 |
| | | | Yes | **0.3438** | **0.4084** | **0.4316** | **0.4414** | **0.4422** |
| | | MLP | No | | 0.3398 | 0.4490 | 0.4738 | 0.5098 |
| | | | Yes | **0.3890** | **0.4292** | **0.4718** | **0.4890** | **0.5154** |
| | | Text2cpts | No | | 0.4012 | 0.4808 | 0.4990 | 0.5058 |
| | | | Yes | **0.3914** | **0.4656** | **0.4958** | **0.5070** | **0.5082** |
| | t2i P@1 | CSA | No | | 0.0168 | 0.1184 | 0.2192 | **0.2404** |
| | | | Yes | **0.1420** | **0.1952** | **0.2330** | **0.2578** | 0.2278 |
| | | MLP | No | | 0.2682 | **0.3586** | **0.3756** | **0.3884** |
| | | | Yes | **0.3808** | **0.3186** | 0.3536 | 0.3646 | 0.3874 |
| | | Text2cpts | No | | 0.2878 | 0.3458 | 0.3574 | 0.3638 |
| | | | Yes | **0.3570** | **0.3584** | **0.3590** | **0.3600** | **0.3662** |

