# OpenReview forum: "Supervised Classification Heads as Semantic Prototypes: Unlocking Vision-Language Alignment via Weight Recycling"
_ICML.cc/2026/Conference — ICML 2026 regular_

### Official Review · Reviewer_cYtc · 2026-03-03

**Soundness:** 3
**Presentation:** 3
**Significance:** 2
**Originality:** 2
**Overall Recommendation:** 4
**Confidence:** 4

**Summary:**

The paper found an interesting method of adapting the linear head from models trained on ImageNet -21K or -1K to the alignment between vision encoders and language models.
The experiments mainly focus on the classification, inlcuding zero-shot, few-shot classification, and retrieval task, mainly following the original experiments setups in the clip paper.

**Compliance With Llm Reviewing Policy:**

Affirmed.

**Final Justification:**

The paper evaluates the method mainly on two tasks, zero-shot classification and retrieval. While the performance for the zero-shot classification lags behind, the performance for the retrieval task shows the positive signal of the method. So would love to improve my rating.

**Key Questions For Authors:**

See above and I would increase the score if I am convinced after rebuttal or other reviewers are positive on the paper. Overall I like this idea.

**Limitations:**

Yes

**Strengths And Weaknesses:**

The idea is simple and effective, which I like it. Aslo if being with the experiment setup in the original clip paper, the experiemnts are solid enough to support the method.

I have one main concern regarding the experimental results: they are consistently below those obtained with CLIP.
The paper claims that CLIP models are trained on roughly 400M image–text pairs, whereas the proposed method requires fewer than a thousand pairs with the linear head augmentation.
But I think one important piece of experiments is missed: when using the linear head augmentation, how many pairs are actually required to reach performance comparable to CLIP?
The expected number of pairs should certainly be far less than 400M.
It might be on the order of hundreds of thousands—for example, similar to how LLaVA is trained using around 500K instruction traces.
So this could convince people to use this method as a training strategy because it can reduce the training cost a lot, and make the linear classification head augmentation as a standard training strategy for the future models.

---

> ### Author Rebuttal · Authors · 2026-03-30
>
> We thank the reviewer for the positive assessment of the paper, for recognizing that the idea is simple and effective, and for noting that the experiments are solid when viewed in the CLIP-style evaluation setup.
>
> > The paper claims that CLIP models are trained on roughly 400M image-text pairs, whereas the proposed method requires fewer than a thousand pairs with the linear head augmentation. But one important experiment is missing: when using the linear head augmentation, how many pairs are actually required to reach performance comparable to CLIP?
>
> Thank you for the suggestion. It is worth noting that our method is not an alignment approach in itself, but rather orthogonal to existing alignment methods, as it augments whatever alignment strategy is used. In this regard, the relevant question is not how many image-text pairs are needed to match CLIP, but how much data is needed to match the performance of any alignment method when it uses only image-text representations, instead of those augmented with classifier weight representations. Concretely, as shown in Table 5, augmenting with recycled classifier weights allows alignment methods to reach a given retrieval performance with significantly less paired data: models trained on just 1K image-caption pairs with weight augmentation outperform the same alignment methods trained on 10K pairs alone, while approaching the performance obtained with 30K pairs. In addition, when pairing weight augmentation with MLP-based alignment, fewer than 1K image-caption pairs are already sufficient to match or surpass CLIP on several benchmarks (CIFAR-10, CIFAR-100, Flowers102), while remaining competitive on others (DTD, Places, Pets). These results validate the core insight of this work: that the classification weights typically discarded after supervised pretraining constitute high-quality semantic anchors.

---

> > ### Author Rebuttal · Reviewer_cYtc · 2026-04-07
> >
> > I have checked the response and also checked the responses for other reviewers.

---

### Official Review · Reviewer_F18t · 2026-03-11

**Soundness:** 3
**Presentation:** 2
**Significance:** 3
**Originality:** 2
**Overall Recommendation:** 4
**Confidence:** 4

**Summary:**

This paper proposes a novel and efficient approach for vision-language alignment by "recycling" the classification head weights from standard supervised pre-trained vision models as semantic prototypes. The core insight is that these normally discarded weights function as high-quality, language-aligned anchors. Leveraging this, the authors introduce a framework that enables post-hoc alignment between independent image and text encoders in two key regimes: achieving alignment with zero image-text pairs, and significantly boosting the performance of existing alignment methods by using the weights as a robust data augmentation source. Experiments across zero/few-shot classification and cross-modal retrieval tasks demonstrate the effectiveness and data efficiency of the proposed "weight recycling" strategy.

**Compliance With Llm Reviewing Policy:**

Affirmed.

**Final Justification:**

The core idea of "weight recycling" is original and the paper demonstrates its practical utility for data-efficient vision-language alignment. While the approach remains partly empirical and its performance is tied to the pre-training label set, its novelty and demonstrated benefits justify publication. I recommend a weak accept (4).

**Key Questions For Authors:**

1 The authors need to clarify precisely how D_weights is used to train the functions g_I and g_T in Equation (2) under different alignment methods (e.g., CSA, text-to-concepts, MLP). Concurrently, the evaluation procedures for different downstream tasks (zero-shot classification, cross-modal retrieval, few-shot classification) are not explicitly described. This ambiguity makes it difficult to assess the rationale behind many experimental setups and comparative baselines. It is recommended to add more detailed algorithmic descriptions, training objective formulas in the methodology section (e.g., Sections 4.1, 4.2), and consider supplementing Figure 2 with illustrative details.

2 The direct efficiency comparison with CLIP in Section 4.2 is inappropriate. This work is a post-hoc alignment​ method, premised on already pre-trained​ image (BEiT) and text (CLIP text) encoders. In contrast, CLIP is an end-to-end jointly pre-trained​ model. This comparison overlooks the massive data and computational resources consumed by the pre-training of BEiT and the CLIP text encoder itself, severely exaggerating the data and computational efficiency advantages claimed in this work. It is suggested to remove this misleading comparison or refocus the discussion on a fair comparison with other post-hoc alignment methods regarding data requirements and computational cost.

3 To prove that "weight representations are better for alignment than image representations," the authors designed the experiment in Table 3, using "average image representations" as the baseline for the image group. This introduces a serious confounding variable: "average image representations" are class-level​ statistical features, while the comparison task (Flickr30K retrieval) is an instance-level​ matching task. Using class-level features to train an instance-level task inevitably leads to performance degradation due to the loss of instance-specific information. This result does not prove that "weights are better"; it only demonstrates that "the averaging operation destroys the effectiveness of image representations for the retrieval task."

**Limitations:**

The authors have partially discussed limitations, noting the modality gap between weights and images and the suboptimal performance on highly specialized datasets. However, the discussion could be more comprehensive. The work is inherently limited by its dependence on the semantic coverage and label quality of the pre-training dataset (e.g., ImageNet-21K), which may not generalize to novel, open-world concepts.

**Strengths And Weaknesses:**

Strengths:

The core idea of repurposing discarded classification head weights as semantic prototypes for vision-language alignment is creative and offers a fresh perspective on leveraging existing supervised models. The demonstration of a "modality gap" between these weights and image features, akin to the text-image gap, is an intriguing empirical finding.

Weaknesses:

Please refer to the Key Questions For Authors.

---

> ### Author Rebuttal · Authors · 2026-03-30
>
> We thank the reviewer for recognizing the creativity of repurposing discarded classifier weights as semantic prototypes for vision-language alignment and for highlighting the modality-gap analysis as an intriguing empirical finding. We also appreciate the three questions on methodological clarity, the positioning of the efficiency claim, and the interpretation of the weight-vs-image ablation. Below we clarify these points.
>
> > 1.
>
> **On $\mathcal{D}_{\text{weights}}$ across alignment methods.** The rationale behind our setups does not depend on the specifics of any particular aligner. A key contribution is that weight recycling is *orthogonal* to the choice of alignment method: despite differing in architecture and loss, CSA, text-to-concepts, and MLP-alignment can all use $\mathcal{D}_{\text{weights}} = \{(w_i, f_T(t_i))\}_{i=1}^{C}$ as training data (all methods are trained with image-text representation pairs), alone or augmented as $\mathcal{D}_{\text{aug}} = \mathcal{D}_{\text{imgtxt}} \cup \mathcal{D}_{\text{weights}}$, by leveraging the compatibility between classifier weights and image representations. This is why we adopted a method-agnostic formulation.
>
> **On implementation and evaluation details.** These details were present but perhaps too spread across the paper: aligner descriptions (CSA relies on Canonical Corr. Analysis, text-to-concepts is a cross-modal mapping, MLP-alignment a two-layer MLP with cosine loss) appear in the main text, benchmark metrics in the appendix, and full implementation details in the released code. To improve accessibility, we have: (1) added an appendix section centralizing details of all alignment methods; (2) noted in Appendix G that classification uses the prompt *"A photo of a [class]"*; and (3) updated Figure 2 to clarify that subfigure (a) depicts a general framework covering all three aligners, with annotations highlighting key differences.
>
> > 2.
>
> We agree that the sentence comparing our results to CLIP can be misleading if interpreted as a comparison of total end-to-end training cost. This is not our intended claim. Our paper is not about competing with CLIP-scale joint pretraining, nor do we advocate training a supervised BEiT-style model solely to obtain classifier weights. Rather, our main contribution is to show that classifier weights, typically discarded after supervised pretraining, behave as meaningful semantic prototypes. This is supported by the cosine-classifier evidence, the semantic-alignment analysis, and the Neural Collapse motivation, while the alignment results are applications of this insight. In particular, once a supervised backbone is already available, its classifier weights come for free as a byproduct of pretraining and can be recycled to unlock vision-language capabilities without image-text pairs at the alignment stage, or to augment paired data thanks to their compatibility with image representations. Following the reviewer's advice, we have rephrased the relevant sentences to make clear that the efficiency discussion concerns only the additional data and computation required for the alignment step, not the original pretraining cost of the encoders.
>
> > 3.
>
> Since both the weight representations and the averaged image representations are class-level, and both are evaluated on the same instance-level Flickr30K retrieval task, any potential mismatch between class-level training and instance-level evaluation applies equally to both approaches. Therefore, we considered the comparison to be fair.
> However, to address the reviewer's concern more directly, we also added an experiment using one randomly sampled image representation per ImageNet-1K class, paired with the corresponding class name and trained with the same alignment protocol. The weights representation alignment still performs substantially better on Flickr30K retrieva.: We included these results in the manuscript.
> | Metric | Weight repr. | Image repr. |
> | --- | ---: | ---: |
> | i2t mAP | 27.51 | 13.11 +/- 0.74 |
> | i2t P@1 | 32.30 | 14.86 +/- 1.52 |
> | i2t P@5 | 23.88 | 11.26 +/- 0.65 |
> | t2i P@1 | 22.12 | 9.42 +/- 0.38 |
>
> As a complementary check, we also evaluated on zero-shot classification benchmarks:
>
> | Dataset | Weight repr. | Image repr. |
> | --- | ---: | ---: |
> | RESISC | 21.16 | 14.19 +/- 2.97 |
> | EUROSAT | 17.70 | 16.18 +/- 2.58 |
> | FLOWERS | 11.82 | 5.90 +/- 2.38 |
> | PETS | 75.69 | 74.50 +/- 1.48 |
> | FOOD | 20.96 | 12.92 +/- 1.14 |
> | CIFAR10 | 89.35 | 87.64 +/- 1.17 |
> | CIFAR100 | 61.12 | 52.78 +/- 1.13 |
> | DTD | 22.71 | 15.43 +/- 1.25 |
> | PLACES | 24.32 | 15.24 +/- 0.91 |
>
> > The work is inherently limited by its dependence on the semantic coverage and label quality of the pre-training dataset (e.g., ImageNet-21K), which may not generalize to novel, open-world concepts.
>
> We do not view this as a limitation specific to our method, but as a general property of current VLMs. Please see the answer submitted to Question 2 from YwCX.

---

> > ### Author Rebuttal · Reviewer_F18t · 2026-04-02
> >
> > Thank you for your response.
> > - In the authors' rebuttal for point 1, some formulas were not fully displayed. Please carefully check and correct the rendering of all equations.
> > - I understand the authors' intention. The original statement regarding data efficiency was inappropriate and could easily be misleading. The authors have now revised it appropriately.
> > - The authors have effectively addressed my concerns through further explanations and additional experiments.

---

> > > ### Author Response · Authors · 2026-04-02
> > >
> > > We are glad that our clarification, additional experiments, and revisions have addressed your concerns. We thank the reviewer for their constructive feedback, which helped us improve the clarity of the paper and strengthen the presentation of our contributions.
> > >
> > > Thank you for pointing out the rendering error, what we intended to write was the following (we now use $D$ instead of $\mathcal{D}$ here for correct rendering):
> > >
> > > **On** $D_{weights}$ **across alignment methods.** The rationale behind our setups does not depend on the specifics of any particular aligner. A key contribution is that weight recycling is orthogonal to the choice of alignment method: despite differing in architecture and loss, CSA, text-to-concepts, and MLP-alignment can all use
> > >
> > > $D_{weights} = \\left(w_i, f_T(t_i)\\right)_{i=1}^{C}$
> > >
> > > as training data, either alone or augmenting image-text pairs as
> > >
> > > $D_{aug} = D_{imgtxt} \\cup D_{weights}$
> > >
> > > by leveraging the compatibility between classifier weights and image representations. This is why we adopted a method-agnostic formulation

---

### Official Review · Reviewer_YwCX · 2026-03-12

**Soundness:** 4
**Presentation:** 4
**Significance:** 3
**Originality:** 4
**Overall Recommendation:** 5
**Confidence:** 3

**Summary:**

The submission proposed a novel method to align the embeddings of pretrained unimodal visual and text encoders without pairwise image-text data. This is achieved by assuming the visual encoder is trained together with a classifier and reusing the weights of the last layer of the classifier as semantic prototypes to be matched with the text embeddings of the class names. The proposed idea is intuitive and shown to be effective by combining visual encoders pre-trained on ImageNet with several text encoders on multiple benchmarks. It not only works independently but also serves as an augmentation to conventional post-hoc image-text alignment solutions that require pairwise data.

**Compliance With Llm Reviewing Policy:**

Affirmed.

**Final Justification:**

I found the ideas proposed in this submission interesting and they are shown effective. I will keep my initial vote of accept.

**Key Questions For Authors:**

1. Does the list of class labels on ImageNet pose a bottleneck to the expressiveness of the projected embeddings? For example, how well do the projections learned on ImageNet generalize to domains that have very different class names?

2. How well does the proposed method scale to different numbers of classes? Can the alignment be learned directly on target domains with limited classes without pre-training on ImageNet? It is also a practical problem that one has a domain-specific image classifier and would like to extend it to enable image retrieval by text. Given that the names of target classes can be significantly different from those on ImageNet, e.g. names of products in a retail store, can the projections be learned solely using the domain-specific classifier weights? How will the number of classes affect the results?

**Limitations:**

Yes

**Strengths And Weaknesses:**

The submission is technically sound and the empirical study is comprehensive. The proposed idea is general and can work with existing post-hoc image-text alignment methods that require pairwise data. It was justified with both empirical evidence and theory. It was evaluated on different settings, including image retrieval, few-shot and zero-shot image classification. The paper is well structured and easy to follow. Details of implementation and experiments are elaborated. The post-hoc image-text alignment task studied here is critical in practice when domain-specific visual encoders are available and need to be aligned with text to enable retrieval, while large-scale pairwise data is hard to collect. This provides a new perspective on this line of research.

---

> ### Author Rebuttal · Authors · 2026-03-30
>
> We thank the reviewer for the positive assessment of the paper, for recognizing its soundness and comprehensiveness, and for acknowledging that the proposed approach is supported by both theoretical justification and empirical evidence. We also appreciate the reviewer's comment that the paper provides a new perspective on this line of research. We address the questions below.
>
> > Does the list of class labels on ImageNet pose a bottleneck to the expressiveness of the projected embeddings? For example, how well do the projections learned on ImageNet generalize to domains that have very different class names?
>
> This is a very interesting question. Our experiments suggest that ImageNet classes do matter, but they are not a hard bottleneck. In particular, we evaluated the method on datasets with relatively low semantic overlap with ImageNet, such as EuroSAT and DTD. As expected, performance is lower than on benchmarks with stronger overlap, such as CIFAR, Flowers, or Pets, but it remains clearly non-trivial. For example, with BEiT-B/16 aligned using only ImageNet-21K classifier weights, we obtain 23.67% acc on EuroSAT and 35.37% on DTD.
> In particular, as we noted in the manuscript:
>
> *…while ImageNet-21K pretraining weight representations may contain semantic overlap with downstream benchmarks, we note that CLIP has also likely seen most of the classes in the downstream classification benchmarks (Xu et al. (2024)). For instance, Xu et al. (2024) reconstructs CLIP's data curation process and finds over 700 out of the 1K classes in ImageNet-1K present in pretraining metadata and observes a correlation between downstream zero-shot classification accuracy and the number of classes matched in the metadata.*
>
> We therefore view semantic overlap as a general factor affecting zero-shot transfer in current vision-language models, rather than a limitation specific to our method. At the same time, our results on lower-overlap datasets indicate that the learned projection still transfers meaningful semantic structure beyond the ImageNet label set.
>
> References:
>  Xu, H. et al. Demystifying CLIP Data. In International Conference on Learning Representations (ICLR), 2024.
>
> > How well does the proposed method scale to different numbers of classes? Can the alignment be learned directly on target domains with limited classes without pre-training on ImageNet? It is also a practical problem that one has a domain-specific image classifier and would like to extend it to enable image retrieval by text. Given that the names of target classes can be significantly different from those on ImageNet, e.g. names of products in a retail store, can the projections be learned solely using the domain-specific classifier weights? How will the number of classes affect the results?
>
> Our evidence suggests that both the number and the diversity of classes matter, and that scale is beneficial. In the paper, using ImageNet-21K weight representations provides substantial improvements over ImageNet-1K, even though ImageNet-1K has some advantages in representation quality: classes are balanced, and are semantically disjoint leaf categories without hypernym-hyponym overlap (e.g. chihuahua and mammal are not two coexisting classes). This suggests that these advantages are outweighed by the approximately 21 times larger number of weight representations in ImageNet-21K, covering a broader region of the latent space and consequently yielding better downstream alignment. We also partially addressed the domain-specific setting in the appendix using iNaturalist weights. There, we found that domain-specific weights can indeed induce non-trivial alignment, but performance is significantly lower than when using ImageNet-21K weights on downstream benchmarks. This suggests that domain-specific classifier weights can be used on their own, but that their effectiveness heavily depends on how broad and diverse the underlying class set is.
>
> For practical settings, such as a retail-store classifier, our current recommendation is to use the domain-specific classifier weights together with whatever image-text pairs are available, rather than relying only on the weights. Our augmentation experiments show that recycled classifier weights are compatible with image-text pairs and provide the largest benefits in the low-data regime. Therefore, this appears to be the most robust strategy for specialized domains with limited classes.

---

> > ### Author Rebuttal · Reviewer_YwCX · 2026-04-04
> >
> > The authors' response addresses my major concern about how well the proposed method might work when the downstream class names are different from the pre-trained ones vastly.

---

### Official Review · Reviewer_BcZk · 2026-03-16

**Soundness:** 2
**Presentation:** 3
**Significance:** 2
**Originality:** 3
**Overall Recommendation:** 3
**Confidence:** 4

**Summary:**

This paper explores whether classification head weights from supervised vision models can be reused as semantic prototypes for vision-language alignment. The authors propose a post‑hoc alignment framework that pairs classifier weights from  ImageNet‑21K with textual class names to construct pseudo alignment data between independently pretrained image and text encoders. The approach can be used either without explicit image-text pairs or to augment existing alignment datasets. The paper presents extensive experiments across zero‑shot classification, cross‑modal retrieval, and few‑shot classification tasks using multiple vision encoders and alignment methods.

**Compliance With Llm Reviewing Policy:**

Affirmed.

**Key Questions For Authors:**

Rather than posing specific questions, I offer the following suggestions that may help strengthen the work in a future revision.

Instead of positioning classifier weight recycling as an alternative to large-scale vision-language training, the idea may be more impactful as a complementary signal for existing multimodal models. For example:

1. **Improving CLIP alignment.** Classifier weights could act as semantic anchors to help mitigate the modality gap observed in vision-language models.
2. **Few-shot adaptation of vision–language models.** The semantic prototypes encoded in classifier weights may provide useful priors for adapting CLIP-like models to new classes with limited labeled data.
3. **Domain adaptation.** Recycling classifier weights from multiple supervised datasets could provide structured semantic knowledge that helps adapt vision-language models to specialized domains.
4. **Data-efficient multimodal training.** Classifier weights could potentially reduce the amount of image-text data required when combined with small multimodal datasets.

Exploring such directions could help clarify the unique advantages of classifier-weight recycling and better demonstrate its practical value.

**Limitations:**

Yes

**Strengths And Weaknesses:**

**Strengths:**

1. **Interesting conceptual perspective.**
The paper introduces a novel viewpoint: classification head weights from supervised vision models can be interpreted as semantic prototypes and potentially reused for vision-language alignment. This idea is conceptually appealing and may inspire new directions for leveraging supervised pretraining artifacts that are typically discarded after training.

2. **Extensive empirical exploration.**
The paper contains a large number of experiments covering zero-shot classification, cross-modal retrieval, and few-shot classification (seen in appendix). The authors also evaluate multiple vision architectures, several alignment approaches, and different text encoders, and include additional analyses examining semantic alignment properties and modality gaps.

3. **Clear presentation and structured narrative.**
The paper is generally well written and logically structured. The motivation, formulation of the post‑hoc alignment framework, and the experimental sections are easy to follow. The figures and tables effectively support the discussion.

4. **Potential usefulness in low-data regimes.**
The idea of using classifier weights as additional semantic anchors could be useful in settings where large image-text datasets are unavailable. The improvements observed in low-data regimes suggest that classifier weights may serve as a complementary signal for existing alignment approaches.

**Weaknesses:**

1. **The work primarily demonstrates feasibility rather than establishing a convincing alignment mechanism.**
The paper shows that classifier weights contain semantic information and can be used to train lightweight alignment models. However, the resulting models remain far from the performance of established vision–language models such as CLIP, and the empirical gains are modest. As a result, the current study mainly illustrates the possibility of using classifier weights as auxiliary signals rather than clearly demonstrating a strong or broadly effective alignment approach.

2. **Strong dependence on CLIP text embeddings weakens the "no image–text pairs" claim.**
Most experiments rely on the CLIP text encoder as the language representation space. CLIP’s text encoder was trained jointly with an image encoder on hundreds of millions of image–text pairs. Table 8 shows that replacing the CLIP text encoder with language-only models (RoBERTa, MPNet, MiniLM) leads to substantially weaker performance. This suggests that much of the cross-modal structure exploited by the proposed method may already be present in CLIP’s visually grounded text embeddings. Consequently, the claim that alignment can be achieved purely from classifier weights without multimodal supervision is somewhat weakened.

Overall, the paper presents an interesting conceptual perspective and a thorough empirical exploration. However, the current evidence is not yet sufficiently convincing that classifier weight recycling provides a robust or broadly applicable mechanism for vision–language alignment. The method appears to rely heavily on visually grounded CLIP text embeddings and the empirical gains remain limited, making the work feel more like an exploratory investigation of a promising idea than a clearly established alignment approach.

---

> ### Author Rebuttal · Authors · 2026-03-30
>
> We thank the reviewer for recognizing the conceptual interest of viewing classifier weights as semantic prototypes, the extensive empirical study, the clear presentation, and the potential usefulness of the approach in low-data settings. Please find our answers below.
>
> ### Key Questions for Authors:
>
> > "Improving CLIP alignment. Classifier weights could act as semantic anchors to help mitigate the modality gap observed in VL models."
>
> As we show in 4.3. Further analyses > Modality Gap..., there is also a modality gap between classifier weights and image representations, which makes using weights as semantic anchors difficult. In Appendix I, our attempts to mitigate this gap did not improve downstream performance. This is also consistent with prior observations that reducing a geometric modality gap does not necessarily improve downstream results (Liang et al., 2022).
>
> [1] Liang et al., Mind the gap: Understanding the modality gap in multi-modal contrastive representation learning. Advances in Neural Information Processing Systems, 2022.
>
> > "Few-shot adaptation of VL models. The semantic prototypes encoded in classifier weights may provide useful priors for adapting CLIP-like models to new classes with limited labeled data."
>
> Our few-shot results already study this. In 4.2. Further analyses > Few-shot classification, even a simple strategy built on these semantic prototypes outperforms strong few-shot classifiers such as NCC in most settings.
>
> > "Domain adaptation. Recycling classifier weights from multiple supervised datasets could provide structured semantic knowledge that helps adapt VL models to specialized domains."
>
> Our paper partially addresses this through the domain-specific weight experiments, but does not explore combining weights from multiple datasets. An important requirement for such a combination to be meaningful is that the weights share a common feature space (e.g., weights are linear heads on top of the same frozen visual encoder). We have added a note on combining weights from multiple datasets and the mentioned requirement in the analysis section, alongside the domain-specific weight experiments.
>
> > "Data-efficient multimodal training. Classifier weights could potentially reduce the amount of image-text data required when combined with small multimodal datasets."
>
> Our retrieval experiments already show that classifier weights can augment small image-caption datasets, especially in the low-data regime (Fig. 4). Most notably, Table 5 in Appendix shows that augmenting just 1K image-caption pairs with recycled weight representations can outperform models trained with 10K image-caption pairs alone (see CSA rows), while approaching the performance obtained with 30K pairs.
>
> ### Weaknesses:
>
> > The work primarily demonstrates feasibility rather than establishing a convincing alignment mechanism
>
> The weight augmentation part of our manuscript is an augmentation strategy (orthogonal to alignment methods) that boosts the performance of existing post-hoc alignment methods in low-data regimes, not an alignment mechanism itself. The appropriate comparison is therefore not against CLIP, but against those same alignment methods without weight augmentation (e.g. Fig. 4, Fig. 6, Tab. 5). That said, even when considering CLIP as a reference point, the results are compelling: combining weight augmentation with a basic yet powerful alignment technique such as entropy alignment, and using fewer than 1K image-text pairs, already yields performance comparable to or surpassing CLIP on several benchmarks (Fig. 4).
>
> > Strong dependence on CLIP text embeddings weakens the 'no image-text pairs' claim.
>
> Our claim is that a supervisedly pretrained image encoder can be endowed with vision-language (VL) capabilities without any paired image-text data in the alignment step. In practice, one may wish to equip a specific supervised pretrained image backbone with text capabilities (e.g., for efficiency or model-size reasons), and our insight on classifier weights as semantic prototypes makes this possible without image-text pairs (which were required by previous methods). Thus, using CLIP's text encoder to endow the image encoder with VL capabilities does not contradict our claim.
> Table 8 shows that alignment is still possible with purely text-only encoders, though performance is lower. Stronger results with CLIP reflect a stronger representation space, not hidden use of image-text pairs for endowing the vision encoder with text capabilities.

---

> > ### Author Rebuttal · Reviewer_BcZk · 2026-04-04
> >
> > I thank the authors for the detailed and thoughtful rebuttal. The responses help clarify the intended positioning of the work, in particular that the primary claim is not to replace large-scale vision–language training, but to demonstrate that a supervised pretrained vision encoder can be endowed with vision–language capabilities without requiring image–text pairs during the alignment step. This clarification is helpful and improves the framing of the contribution.
> >
> > That said, my concern remains on the practical relevance of the proposed approach within this setting. While the rebuttal motivates scenarios where one may wish to equip an existing supervised vision backbone with language capabilities in a data and compute-efficient manner, it remains somewhat unclear how the proposed method compares to practical alternatives in such scenarios (e.g., lightweight adaptation of existing vision-language models, prompt tuning, or linear probing), particularly given its reliance on CLIP text representations for strong performance. While the paper demonstrates the potential of classifier weights as semantic prototypes, comparisons against such alternatives would be important to better establish the relevance and practical utility of the approach.

---

> > > ### Author Response · Authors · 2026-04-04
> > >
> > > We thank the reviewer for raising this important point. We would like to clarify that this question is not specific to our proposed weight augmentation, but applies more broadly to the motivation behind post-hoc alignment methods. In particular, the goal of such methods (e.g. cited work such as CSA) is to endow a supervised vision encoder with text capabilities, rather than adapting an existing vision–language model. This raises the natural question of when this paradigm is preferable to alternatives such as prompt tuning or lightweight adaptation of pretrained VLMs.
> > >
> > > A key distinction from these alternatives (prompt tuning, lightweight adaptation) is that they require adopting the vision backbone of a pretrained VLM (typically ViT-based architectures in CLIP-like models). In contrast, post-hoc alignment methods operate on arbitrary vision encoders, without requiring any modification of their architecture.
> > >
> > > This is particularly relevant in practical scenarios where a specific vision backbone must be retained. For example, one may wish to use an EfficientNet model pretrained on ImageNet-21K, either for efficiency reasons, deployment constraints, or compatibility with existing interpretability tools tailored to that architecture. In such cases, adapting an existing VLM (e.g., via prompt tuning on a ViT-based CLIP model) does not transfer the desired vision architecture, and therefore cannot directly address this need.
> > >
> > > Our method builds on this paradigm by showing that classifier weights can serve as an additional semantic signal to improve such post-hoc alignment, particularly in low-data regimes.

---

### Decision · Program_Chairs · 2026-04-30

**Decision:**

Accept (regular)

**Comment:**

This paper presents an approach for post-hoc vision-language alignment by reusing the classification heads of a pretrained encoder to support vision-language alignment without paired data.  This paper was reviewed by four experts where 3 of 4 noted the paper's potential benefits.  The one reviewer who argued for rejection did so, in part, by noting that the applications of this work are unclear (i.e., those proposed are not fully convincing).  However, the authors argue that this would be a collective failing of the all the work on post-hoc vision-language alignment.  To put another way, the benefits of this topic have been adjugated in prior work who have judged this work as valuable.  While the AC finds this argument has some merit, due to the accepted agreement of both the other reviewers in this paper and those in prior work,  it is not sufficient to overturn the majority recommendation to acceptance.  The authors are encouraged to ensure reviewer comments are addressed when preparing their camera ready.